

# A new genus of oryzomyine rodents (Cricetidae, Sigmodontinae) with three new species from montane cloud forests, western Andean cordillera of Colombia and Ecuador

Jorge Brito[1], Claudia Koch[2], Alexandre R. Percequillo[3], Nicolás Tinoco[4], Marcelo Weksler[5], C. Miguel Pinto[6] and Ulyses F. J. Pardiñas[1,7]

[1] Instituto Nacional de Biodiversidad (INABIO), Quito, Ecuador
[2] Zoologisches Forschungsmuseum Alexander Koenig (ZFMK), Bonn, Germany
[3] Departamento de Ciências Biológicas, Escola Superior de Agricultura "Luiz de Queiroz", Universidade de São Paulo, Piracicaba, São Paulo, Brazil
[4] Sección de Mastozoología, Museo de Zoología, Facultad de Ciencias Exactas y Naturales, Pontificia Universidad Católica del Ecuador, Quito, Ecuador
[5] Setor de Mastozoologia, Departamento de Vertebrados, Museu Nacional, Universidade Federal do Rio de Janeiro, Rio de Janeiro, Brazil
[6] Observatorio de Biodiversidad Ambiente y Salud (OBBAS), Quito, Ecuador
[7] Instituto de Diversidad y Evolución Austral (IDEAus—CONICET), Puerto Madryn, Argentina

Corresponding author
Jorge Brito, jorgeyakuma@yahoo.es

## ABSTRACT

The Andean cloud forests of western Colombia and Ecuador are home to several endemic mammals; members of the Oryzomyini, the largest Sigmodontinae tribe, are extensively represented in the region. However, our knowledge about this diversity is still incomplete, as evidenced by several new taxa that have been described in recent years. Extensive field work in two protected areas enclosing remnants of Chocó montane forest recovered a high diversity of small mammals. Among them, a medium-sized oryzomyine is here described as a new genus having at least three new species, two of them are named and diagnosed. Although externally similar to members of the genera *Nephelomys* and *Tanyuromys*, the new genus has a unique molar pattern within the tribe, being characterized by a noticeable degree of hypsodonty, simplification, lamination, and third molar compression. A phylogeny based on a combination of molecular markers, including nuclear and mitochondrial genes, and morphological data recovered the new genus as sister to *Mindomys*, and sequentially to *Nephelomys*. The new genus seems to be another example of a sigmodontine rodent unique to the Chocó biogeographic region. Its type species inhabits cloud forest between 1,600 and 2,300 m in northernmost Ecuador (Carchi Province); a second species is restricted to lower montane forest, 1,200 m, in northern Ecuador (Imbabura Province); a third putative species, here highlighted exclusively by molecular evidence from one immature specimen, is recorded in the montane forest of Reserva Otonga, northern Ecuador (Cotopaxi Province). Finally, the new genus is also recorded in southernmost Colombia (Nariño Department), probably represented there also by a new species. These species are spatially separated by deep river canyons through Andean forests, resulting in marked environmental discontinuities. Unfortunately, Colombian and Ecuadorian

Pacific cloud forests are under rapid anthropic transformation. Although the populations of the type species are moderately abundant and occur in protected areas, the other two persist in threatened forest fragments.

## INTRODUCTION

The Oryzomyini is the largest tribe among the 56 extant sigmodontine rodent clades, and according to current counts, it comprises about 152 living (including 57 historical extinct) species distributed in 33 genera (*Weksler, 2015*; *Pardiñas et al., 2017*). It is also the tribe with the widest geographic distribution, extending from the southeastern United States of America to Tierra del Fuego and the Cape Horn islands, plus some oceanic islands and the Antillean region (*Weksler, 2006*; *Pardiñas et al., 2017*).

An important portion of this noteworthy diversity is associated with the Andean slopes of northern South America (trans-Andean and Andean distribution categories sensu *Weksler, 2006*: 83). Several authors, using different methodologies and concepts, identified these regions as major centres of oryzomyine species richness (*Reig, 1984*, *1986*; *Musser et al., 1998*; *Valencia-Pacheco et al., 2011*; *Pine, Timm & Weksler, 2012*; *Prado & Percequillo, 2013*; *Prado et al., 2015*; *Patton, Pardiñas & D'Elía, 2015*; *Maestri & Patterson, 2016*).

The Chocó biogeographic region is one of the zones with the greatest biodiversity and endemism on the planet (*Myers et al., 2000*). For this reason, and because of its high degree of threats to biodiversity, it is considered as one of the 25 Priority Terrestrial Ecoregions of the World and an endemism hotspot (*Mittermeier et al., 1999*; *Myers et al., 2000*). This region comprises westernmost Panamá, Colombia and Ecuador and northernmost western Peru. Of the original 260,660 Km$^2$ only 24% (65.000 Km$^2$) currently remains as native forest (*Brooks et al., 2002*). Current threats to biodiversity of this region include climatic change, the advance of human colonization and infrastructure development, and the direct transformation of the land into agricultural fields. In addition, hunting is a problem for several species of birds and mammals (*Mittermeier et al., 1999*; *Brooks et al., 2002*).

Chocó forests are home to a variety of endemic oryzomyines, ranging from suprageneric clades, such as the *Sigmodontomys-Tanyuromys-Melanomys* clade (*Pine, Timm & Weksler, 2012*), to "*Handleyomys*", *Nephelomys*, *Mindomys*, *Transandinomys* (e.g., *H. alfaroi, N. moerex, N. devius, M. hammondi, T. bolivaris and T. talamancae*). Important elements of this trans-Andean oryzomyine radiation are species of *Transandinomys* and "*Handleyomys*" (*Musser et al., 1998*; *Almendra et al., 2018*), which occupy lowland and montane forests of the Chocó. Despite that, our knowledge of sigmodontine biodiversity of this hotspot is still incomplete. A recent example is the recognition of a new species of *Tanyuromys, T. thomasleei Timm, Pine & Hanson, 2018*. In the montane cloud forests of the Chocó also occurs the poorly-known *Mindomys hammondi*

(*Thomas, 1913*), one of the most enigmatic rodent taxa of South America. *Mindomys* is a monotypic genus with uncertain phylogenetic position (*Weksler, 2006*; *Ronez et al., 2020b*), restricted to Ecuadorean forests between Mindo and Alto Tambo (*Thomas, 1913*; *Weksler, Percequillo & Voss, 2006*; *Percequillo, 2015*; *Pinto et al., 2018*).

One of the major obstacles to our knowledge of Chocó biodiversity is the lack of proper sampling in the region. During the last few years, numerous field expeditions were conducted by the senior author (JB) to assess small mammal assemblages in several sites in northwest Ecuador. As a result, a rich collection of sigmodontine rodents was secured, including at least 20 species (*Brito & Arguero, 2016*; *Curay, Romero & Brito, 2019*). A primary morphological sorting of this material suggested the occurrence of undescribed oryzomyine taxa that, although externally similar to *Nephelomys* and *Tanyuromys*, displayed trenchant differences. These results were confirmed by further morphological and molecular analyses, and by the discovery of additional museum material. The goal of this contribution is to provide the description of these new taxa, representing a new genus and two new species of the tribe Oryzomyini, including phylogenetic relationships determined by morphological comparisons and detailed anatomical evidence, partially based on micro-computed tomography (micro-CT). These new cricetids are added to the endemic list of rodents that inhabit the Chocó montane cloud forests of western Andes in Ecuador and Colombia.

## MATERIALS AND METHODS

### Studied specimens

Specimens representing the new genus described here were mostly obtained from field expeditions conducted by JB and his team in two Ecuadorian protected areas, Reserva Río Manduriacu and Reserva Drácula. The former reserve was sampled during three consecutive nights in April 2017 and September 2019; the latter was surveyed during 18 nights between June 2016 and September 2019. In both places, pitfall traps were employed (Supplemental Information S1), with 10–12 buckets (between 20 and 60 l of capacity) distributed along an 80–120 m drift line, with a total trap effort of 320 trap nights. The pitfall traps were placed near runways, holes, and other signs of small mammal activity, and baited with rolled oats mixed with vanilla and alternating with balanced feed for cows. Handling and all activities regarding specimens followed care and use ethical procedures recommended by the American Society of Mammalogists (*Sikes et al., 2016*). For the use and care of animals, we follow the guidelines of the Ministry of the Environment of Ecuador, through scientific research authorization No 006-2015-IC-FLO-FAU-DPAC MAE and No 003-2019-IC-FLO-FAU-DPAC/MAE. Most of the animals were recovered dead, due to the huge amount of rainwater accumulated in the buckets, despite efforts to drain the water daily (during sampling there were heavy downpour rains; the mean annual precipitation in this region surpasses 3,000 mm). Obtained museum study skins, skeletons, fluid-preserved bodies, and tissue samples stored in 96% ethanol were deposited in the biological collections of the Instituto Nacional de Biodiversidad (INABIO; Quito, Ecuador) and the Departamento de Biología de la Escuela Politécnica Nacional (MEPN; Quito, Ecuador). In addition, one further specimen belonging to the

new genus was originally collected by CMP and deposited in the Museo de Zoología de la Pontificia Universidad Católica del Ecuador (QCAZ). Finally, two Colombian specimens are housed at the Instituto de Ciencias Naturales (ICN; Universidad Nacional de Colombia, Bogotá). As comparative materials we employed specimens of *Mindomys hammondi*, including those of the type series housed at The Natural History Museum (BMNH; London, UK), specimens housed at the Royal Ontario Museum (ROM; Toronto, Canada), the Zoologisches Forschungsmuseum Alexander Koenig (ZFMK; Bonn, Germany), and at the Museum of Zoology, University of Michigan (UMMZ; Ann Arbor, MI, USA). We also inspected series of the genera *Nephelomys* and *Tanyuromys* from Ecuador. All examined specimens are listed in the Supplemental Information S2.

## Anatomy, age criteria and measurements

To describe cranial anatomy, we followed the criteria and nomenclature established by *Hershkovitz (1962)*, *Voss (1988)*, *Carleton & Musser (1989)*, *Steppan (1995)*, *Martinez et al. (2018)* and *Wible & Shelley (2020)*. Molar occlusal morphology was assessed based on *Reig (1977)*, and stomach gross morphology was interpreted according to *Carleton (1973)*. We followed the terminology and definitions employed by *Tribe (1996)* and *Costa et al. (2011)* for age classes and restricted the term "adults" for those in categories 3 and 4. We obtained the following external measurements in millimetres (mm), some of them registered in the field and reported from specimen tags, others recorded in museum cabinets: HB (head and body length), TL (tail length), HF (hind foot length, including claw), E (ear length), LMV (length of longest mystacial vibrissae), LSV (length of longest superciliary vibrissae), LGV (length of longest genal vibrissae), and W (body mass, in grams). Cranial measurements were obtained with digital callipers, to the nearest 0.01 mm. We employed the following dimensions (see *Voss, 1988*; *Brandt & Pessôa, 1994*; and *Musser et al., 1998* for illustrations): ONL (occipitonasal length), CIL (condylo-incisive length), LD (length of upper diastema), LUM (crown length of maxillary toothrow), LIF (length of incisive foramen), BIF (breadth of incisive foramen), BM1 (breadth of M1), BR (breadth of rostrum), LN (length of nasals), LPB (length of palatal bridge), BBP (breadth of bony palate), LIB (least interorbital breadth), ZB (zygomatic breadth), BZP (breadth of zygomatic plate), LB (lambdoidal breadth), OFL (orbital fossa length), BB (bular breadth), LM (length of mandible), LLM (crown length of mandibular toothrow), and LLD (length of lower diastema). Finally, dental measurements, the maximum length and width of each individual molar, were obtained under magnification using a reticulate eyepiece.

## Scanning

To improve the anatomical scrutiny, and also to appreciate the morphology of internal bony structures, the skulls of the holotypes of the two new species (MECN 5928, MEPN 12605) described herein were scanned with a high-resolution micro-computed tomography (micro-CT) desktop device (Bruker SkyScan 1173, Kontich, Belgium) at the ZFMK. To avoid movements during scanning, the skulls were placed in a small plastic container embedded in cotton wool. Acquisition parameters comprised: An X-ray beam

(source voltage 43 kV and current 114 µA) without the use of a filter; 1,200 projections of 500 ms exposure time each with a frame averaging of five recorded over a 360° continuous rotation, resulting in a scan duration of 1 h 13 min; a magnification setup generating data with an isotropic voxel size of 15.97 µm (MEPN 12605) and 17.04 µm (MECN 5928), respectively. The CT-dataset was reconstructed with N-Recon software (Bruker MicroCT, Kontich, Belgium) and rendered in three dimensions using CTVox for Windows 64 bits version 2.6 (Bruker MicroCT, Kontich, Belgium). For comparison, the holotype of *Mindomys hammondi* (BMNH 13.10.24.58) was characterized at the Imaging Analysis Centre of the BMNH using a Nikon Metrology HMX ST 225 (Nikon, Tring, UK). Acquisition parameters comprised: An X-ray beam (source voltage 100 kV and current 150 µA) filtered with 0.1 mm of copper; 3,142 projections of 500 ms exposure time each with a frame averaging of 2 recorded over a 360° continuous rotation; a magnification setup generating data with an isotropic voxel size of 22.67 µm. A filtered back projection algorithm was used for the tomographic reconstruction, using the CT-agent software (Nikon Metrology GmbH, Alzenau, Germany), producing a 16-bit uncompressed raw volume. Finally, this dataset was rendered in three dimensions with Amira software (Thermo Fisher Scientific, Hillsboro, OR, USA).

## Statistics

Females and males were combined in all analyses following *Voss (1991)* and *Abreu-Junior et al. (2012)*, who concluded that sexual dimorphism was not an important source of morphometric variation in oryzomyine rodents. Main univariate descriptive statistics were calculated for the two species described here. We also compared adults using a principal component analysis (PCA) based on log (natural)-transformed cranial measurements and the covariance matrix. PCAs were performed on two subsets of basic data in order to allow the inclusion of different specimens. In an approach focused on Ecuadorian specimens from Drácula and Río Manduriacu samples, we worked on a matrix including six external (HB, TL, HF, E, LMV, LGV), 19 cranial (ONL, CIL, LD, LUM, LIF, BIF, BM1, BR, LN, LPB, BBP, LIB, ZB, BZP, OFL, BB, LM, LLM, LLD), and 12 dental (LM1, WM1, LM2, WM2, LM3, WM3, Lm1, Wm1, Lm2, Wm2, Lm3, Wm3) dimensions. To maximize the geographic coverage including Colombian specimens, we worked on a subset composed of three external (HB, TL, HF), and 12 cranial (CIL, LD, LUM, BIF, BM1, BR, LN, LPB, LIB, ZB, BZP, OFL) dimensions. Statistical procedures were carried out using the software Statistica and PAST (PAleontologicalSTatistics), version3.21 (*Hammer, 1999*).

## DNA amplification and sequencing

DNA extraction was made from liver or muscle samples preserved in 90% ethanol, and from samples taken from museum specimens preserved in 70% ethanol, or as dry skin specimens. In the case of the fluid specimens, samples of muscle or part of the tragus of the ear were taken. Samples of a hind paw or part of the tragus of the ear were taken from dry skin specimens. These samples were subjected to a washing of salts and buffers to eliminate residues that may affect extraction or PCR. For fresh tissue samples

(90% ethanol), DNeasy (Qiagen) or Puregene (Gentra) extraction kits were used. For museum samples the protocol of *Bilton & Jaarola (1996)* was used. We amplified two genes: the mitochondrial cytochrome-b (Cytb) gene using the protocol and primers of *Arellano, Gonzáles-Cózalt & Rogers (2006)*, and the nuclear interphotoreceptor retinoid binding protein (IRBP) gene using the protocol and primers described in *Jansa & Voss (2000)*. The amplicons were sequenced by the company Macrogen (South Korea, Inc). The sequences were edited and assembled using the software Geneious R11 (https://www.geneious.com) and aligned using the Clustal-W tool.

## Morphologic analysis

Oryzomyines were scored for the characters described by *Weksler (2006)* and *Percequillo, Weksler & Costa (2011)*, and employed in previous analyses of oryzomyines (*Voss & Weksler, 2009*; *Pine, Timm & Weksler, 2012*; *Turvey et al., 2010*; *Turvey, Brace & Weksler, 2012*; *Ronez et al., 2020b*). The taxonomic sampling of the morphological matrix corresponds to that of *Pine, Timm & Weksler (2012)* with the addition of the new material described here (Supplemental Information S3). We employed the "polymorphic" coding of *Wiens (1995)* for characters with intraspecific variation, and some characters were treated as ordered, following *Weksler (2006)*. The morphological character matrix constructed for the analyses is provided as Supplemental Information S3, with some modifications on characters referring to number of mammae (characters 1–3), interorbital region (24–26), and braincase (28).

## Phylogenetic analyses

We conducted phylogenetic analyses using two datasets: a total evidence matrix combining morphological characters with the molecular data (Cytb and IRBP), with only one terminal per taxon; a molecular-only analysis (Cytb, IRBP and Cytb + IRBP) using the taxon sampling of *Weksler (2006)* plus new taxa (13 new sequences from Cytb, and 7 new sequences from IRPB, obtained in this study). The phylogenetic trees of the first two datasets were rooted using the neotomine *Peromyscus maniculatus* and the tylomyine *Nyctomys sumichrasti*.

The concatenated morphological and DNA matrix was subjected to phylogenetic analyses using maximum parsimony (MP; *Farris, 1983*; *Swofford et al., 1996*), maximum likelihood (ML; *Felsenstein, 1981*), and Bayesian inference (BI; *Huelsenbeck et al., 2001*; *Yang & Rannala, 1997*), while the molecular datasets were analysed with ML and BI. See Supplemental Information S4 for GenBank accession number, voucher specimens of analysed material and sources of sequences. The heuristic search algorithm implemented by PAUP* version 4.0a166 (*Swofford, 2002*) was used in parsimony analyses. Each heuristic search employed 1,000 replicates of random taxon addition with TBR branch swapping; clades with at least one unambiguous synapomorphy were the only ones retained. Jackknife support values (*Farris et al., 1996*) for the parsimony analyses were calculated using 1,000 pseudoreplicates, with heuristic searches employed within each replicate (36.8% character removal per replicate; 10 random addition replicates, TBR branch swapping, no more than 100 trees saved per replicate).

The evolutionary models for Cytb, IRBP and concatenated genes were obtained with PartitionFinder (*Lanfear et al., 2012*). The models for Cytb were: first position GTR + I + G, second position HKY + G, and third position GTR + I + G; for IRBP were: first position HKY + G, second position and K80 + G and third position K80 + G, and GTR + G + I for all partitions of Cytb and K80 + G for all partitions of IRBP. The parsimony model of *Lewis (2001)* was used for the morphological characters. The maximum-likelihood trees were calculated using RAxML (*Stamatakis, 2006*). Bayesian analyses were performed using Markov Chain Monte Carlo sampling as implemented in Mr Bayes 3.1 (*Huelsenbeck & Ronquist, 2001*; *Ronquist & Huelsenbeck, 2003*). Uniform interval priors were assumed for all parameters except base composition, for which we assumed a Dirichlet prior. We performed four independent runs of 10,000,000 generations each, with two heated chains sampling for trees and parameters every 10,000 generations. The first 2,500,000 generations were discarded as burn-in, and the remaining trees were used to estimate posterior probabilities for each node. All analyses were checked for convergence by the effective sample size (ESS ≥ 500), and the potential scale reduction factor was also verified (PSRF = 1). Nodal bootstrap values for the likelihood analysis were calculated using 1,000 pseudoreplicates, under the GTRCAT model in RAxML (*Felsenstein, 1985*; *Stamatakis, 2006*). Phylogenetic analyses were run in the CIPRES Science Gateway (*Miller, Pfeiffer & Schwartz, 2010*).

## Genetic distances and saturation analysis

The uncorrected genetic p-distances were calculated using the Mega 7 program (*Kumar, Stecher & Tamura, 2016*), the comparisons were made at different taxonomic levels: among the genera most related to the new genus that we describe (*Euroryzomy*, *Hylaeamys*, *Handleyomys*, *Nephelomys*, *Oecomys* and *Transandinomys*) and among the three new species. The matrix includes sequences from 796 bp to 1,140 bp (Supplemental Intormation S3). To explore the degree of saturation, we performed a saturation analysis in the DAMBE6 (*Xia, 2017*) program where we plotted the divergence of the sequences against the number of substitution (transitions and transversions), for each gene partition and for each codon position.

## New zoological taxonomic names

The electronic version of this article in Portable Document Format (PDF) will represent a published work according to the International Commission on Zoological Nomenclature (ICZN), and hence the new names contained in the electronic version are effectively published under that Code from the electronic edition alone. This published work and the nomenclatural acts it contains have been registered in ZooBank, the online registration system for the ICZN. The ZooBank LSIDs (Life Science Identifiers) can be resolved and the associated information viewed through any standard web browser by appending the LSID to the prefix http://zoobank.org/. The LSID for this publication is: urn:lsid: zoobank.org:pub:3E11AF88-BD56-40BE-9D43-EF6E5998E2D1. The online version of this work is archived and available from the following digital repositories: PeerJ, PubMed Central and CLOCKSS.

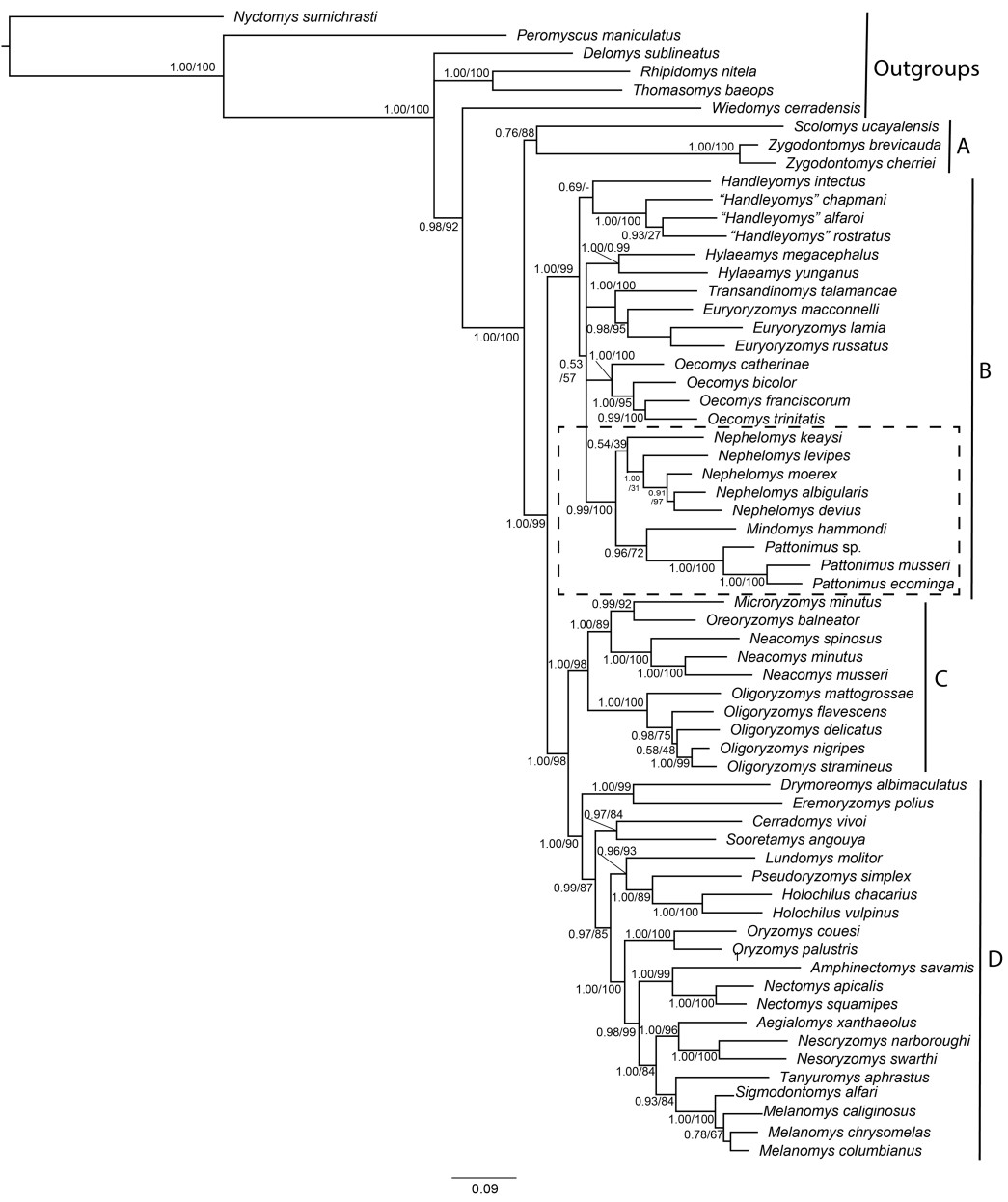

**Figure 1 Phylogenetic relationships of the tribe Oryzomyini.** Phylogenetic relationships of the tribe Oryzomyini. Tree obtained from IB of DNA sequences of mitochondrial (Cytb 800 to 1,143 bp), and nuclear (IRBP 700 to 1,266 bp) genes and 103 morphological characters, from 63 terminals. Numbers below branches are ML bootstrap support and posterior probability values. Letters (A–D) indicate clades. The dash line indicated the position of new taxa.  

## RESULTS

### Phylogeny

The combined matrix of morphological and molecular datasets included 1,126 variables and 818 parsimony-informative characters, of which 95 were morphological characters, 491 from Cytb, and 232 from IRBP. Phylogenetic trees produced by maximum likelihood and Bayesian analyses of this supermatrix were similar (Fig. 1), with high proportions

of nodes with high support, that is, bootstrap support (BS) > 85% and posterior probability (PP) > 0.95. These trees are also similar to previous phylogenetic results for Oryzomyini (*Percequillo, Weksler & Costa, 2011*; *Pine, Timm & Weksler, 2012*; *Turvey et al., 2010*; *Voss & Weksler, 2009*; *Weksler, 2003*, *2006*), with the tribe reconfirmed as monophyletic, and four major clades consistently recovered (clades A to D of *Weksler, 2006*). Clades B, C, and D have high nodal support (BS > 90% and PP = 1), and clade A (containing *Scolomys* and *Zygodontomys*) has a lower support (BS = 88% and PP = 0.77). The topological base of Oryzomyini remains unchanged from previous analyses, with clade C (*Oreoryzomys*, *Neacomys*, *Microryzomys*, and *Oligoryzomys*) representing the sister group to clade D (*Holochilus*, *Pseudoryzomys*, *Oryzomys*, *Nectomys*, *Amphinectomys*, *Aegialomys*, *Nesoryzomys*, *Melanomys*, *Sigmodontomys*, *Tanyuromys*, *Eremoryzomys*, *Drymoreomys*, *Cerradomys*, *Sooretamys*, and *Lundomys*) with high nodal support (BS = 98%, PP = 1). Clade B (*Transandinomys*, *Euryoryzomys*, *Nephelomys*, *Oecomys*, *Hylaeamys*, *Handleyomys*, *Mindomys*), including the new taxa, represented by specimens from Reserva Drácula (sp. 1), and Reserva Río Manduriacu (sp. 2), as well as a specimen from Reserva La Otonga (sp. 3), is sister to clade C + D with high nodal support (BS = 99%, PP = 1). Most intergeneric relationships within clades C and D have high nodal support, but intergeneric relationships within clade B are still poorly supported. Nevertheless, a clade containing *Nephelomys*, *Mindomys* and the new taxa was recovered with high support (BS = 100%, PP = 0.99); within this clade, *Mindomys* was constantly recovered as the sister species to the clade formed by the three new taxa described here, albeit with medium support (BS = 72%, PP = 0.96). The only notable differences between Maximum likelihood and Bayesian inferences include the non-recovery of *Handleyomys* as a monophyletic group in the former, and of *Melanomys* in the latter; these two differences, however, involve relationships with low nodal support.

Parsimony analysis of the supermatrix resulted in four most parsimonious trees (6,967 steps, CI = 0.21, RI = 0.59), the strict consensus of which showed a few changes compared to the structure of trees as recovered in the ML and BI analyses; clades C and D were not recovered as monophyletic, with *Oligoryzomys* not clustering with *Oreoryzomys*, *Microryzomys*, and *Neacomys*, and *Eremoryzomys* and *Drymoreomys* not recovered within clade D. As also described in *Pine, Timm & Weksler (2012)*, this structure is probably due to the phylogenetic signal saturation of the mitochondrial Cytb in higher-level relationships within Oryzomyini in the parsimony analysis (*Weksler, 2003*), which does not correct for multiple substitutions. Results for saturation analysis in DAMBE corroborate this, as cytochrome-b was found to be saturated in all its codon positions (Supplemental Information S6). In any case, clade B was recovered as monophyletic, and within it a clade containing *Mindomys*, *Nephelomys* and the new taxa was also recovered with high support (BS = 87%); in addition, *Mindomys* and the new taxa were found as sister taxa (BS = 86%).

Phylogenetic analyses of the expanded molecular-only matrix (Cytb + IRBP) (Fig. 2), also recovered the new taxa as a monophyletic group in clade B (sensu *Weksler, 2006*); nevertheless, the new taxa were nested within a paraphyletic *Nephelomys*. The clade of the new taxa was sister to *Nephelomys levipes*, and in turn this clade sister to

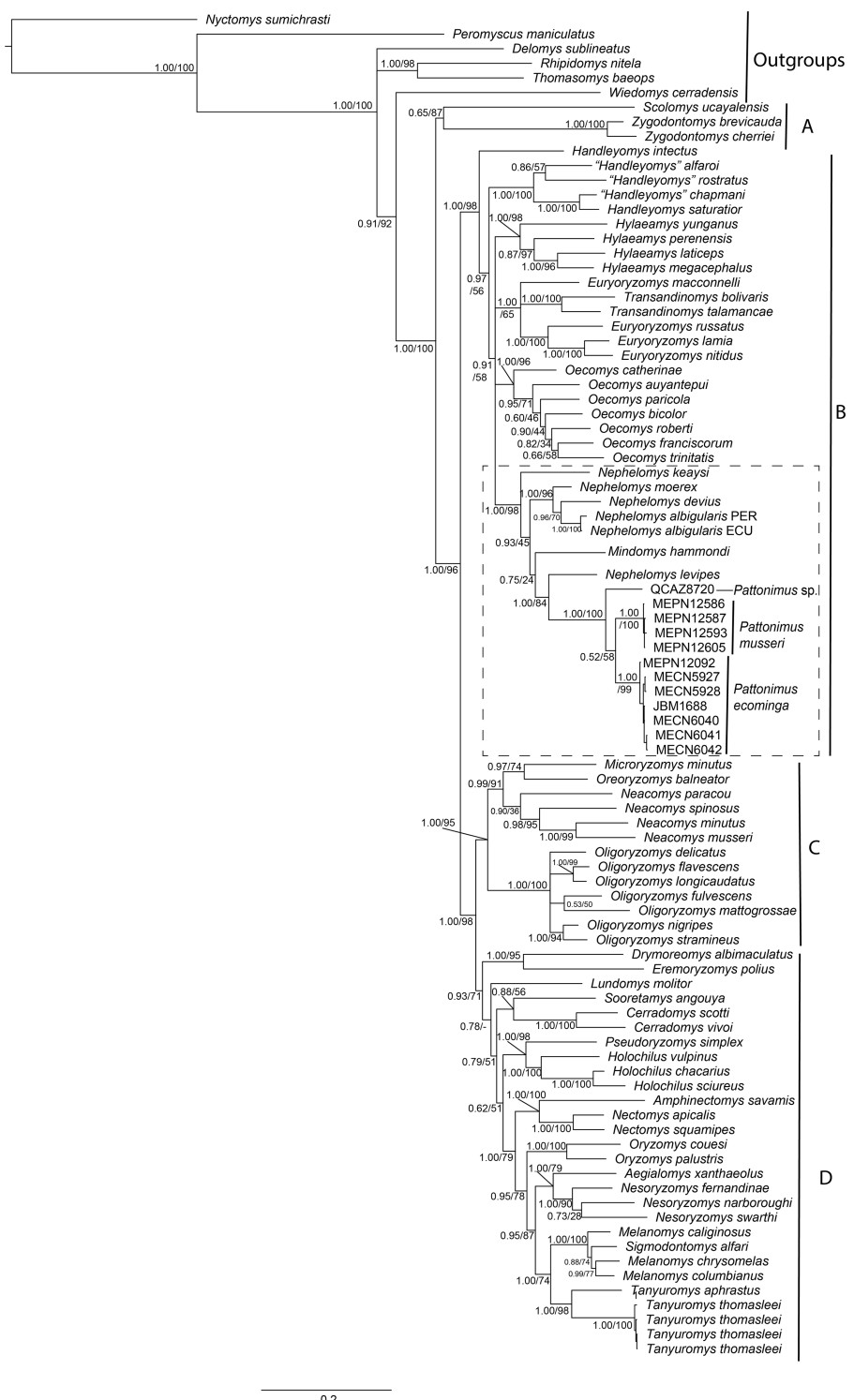

**Figure 2 Phylogenetic relationships of the tribe Oryzomyini.** Phylogenetic relationships of the tribe Oryzomyini. Tree obtained from IB analysis of DNA sequences concatenated Cytb + IRPB genes of up to 2,049 bp. Numbers below branches are bootstrap support and posterior probability values. Letters (A–D) indicate clades. The dash line indicated the position of new taxa.

*Mindomys hammondi*. Nodal support for the clade with the new taxa is high, as well as for the clade including the paraphyletic *Nephelomys*, *Mindomys* and the new taxa; the support for the internal clades are low to moderate, including the clades containing *Mindomys* and *N. levipes* + new taxa.

The phylogenetic analyses of each gene separately presented different topologies, in the case of the Cytb gene (Supplemental Information S5A) *Mindomys* was located nested in *Nephelomys*, while this genus was paraphyletic. The samples of the new species described here formed a monophyletic clade sister to *Nephelomys* + *Mindomys*, and the samples from Reserva Otonga (sp. 3) and Reserva Dracula (sp. 1) formed a clade with high support, and this in turn, it was recovered as sister group to the clade with samples from the Reserva Rio Manduriacu (sp. 3). The IRBP gene (Supplemental Information S5B) brought *Mindomys* back as the sister taxon to a clade formed by *Nephelomys* and the new species described here. Within this clade, *Nephelomys* was paraphyletic since *N. keaysi* and *N. levipes* formed a clade with the samples of the new species described here (Supplemental Information S5B). In the case of the new species, the samples from Reserva Dracula (sp. 1) and Reserva Rio Manduriacu (sp. 2) formed a clade with high support, and it was recovered as sister group to the samples from the Reserva Otonga (sp. 3).

The saturation results showed that the mitochondrial gene Cytb presents a degree of saturation both in the first and second position of the codon, while, in the third position, a high degree of saturation (Supplemental Informations 6A–6C). In the case of the IRBP nuclear gene, it did not present any degree of saturation in the three positions of the codon. The saturation found in the Cytb gene may explain the low resolution in the relationships at the species level (Supplemental Information S6A) as is the case *Nephelomys*, *Mindomys* and the new genus described here, while the IRBP gene presented a better resolution among the species of the genera mentioned above (Supplemental Information S6B).

The levels of genetic differentiation (Table 1) of this new genus with respect to the genera integrating clade B (*Weksler, 2006*) ranged from 11.91% (*Nephelomys*) to 15.64% (*Hylaeamys*). Intrageneric distances among samples from Reserva Drácula (sp. 1), Reserva Río Manduriacu (sp. 2), and Reserva Otonga (sp. 3), were approximately 7% (sp. 1 vs sp. 2 = 7.87% ± 0.87; sp. 1 vs sp. 3 = 7.55% ± 0.83; sp. 2 vs sp. 3 = 7.28% ± 0.96).

## Morphological comparisons

In this section, we compare the new genus with both the phylogenetically closer lineages *Mindomys* and *Nephelomys*, and the geographically closer genus *Tanyuromys* (see Table 2).

Specimens of *Mindomys* exhibit a large body size (HB range: 173–293 mm), while body sizes of adult specimens of the new genus, its sister taxon, are smaller (115–140 mm), as are HB ranges *Tanyuromys* (150–142 mm), and *Nephelomys* (100–228 mm). The tail is very long in individuals of all taxa of this clade, surpassing the HB length: *Mindomys* (TL > 222 mm), *Nephelomys* (TL range: 102–253 mm), and the new genus (TL range: 180–184 mm). Specimens of *Nephelomys* have much more sharply bicolored tails than individuals of the new genus, which lack distinct countershading and have
**Table 1 Uncorrected genetic distances (p distances).**

| | | 1 | 2 | 3 | 4 | 5 | 6 | 7 | 8 |
|---|---|---|---|---|---|---|---|---|---|
| **Among Genus** | | | | | | | | | |
| 1 | *Euryoryzomys* | | 0.86 | 0.76 | 0.88 | 0.79 | 0.77 | 0.91 | 0.83 |
| 2 | *Handleyomys* | 14.01 | | 0.67 | 0.81 | 0.75 | 0.72 | 0.91 | 0.70 |
| 3 | *Hylaeamys* | 13.34 | 14.58 | | 0.79 | 0.76 | 0.66 | 0.92 | 0.77 |
| 4 | *Mindomys* | 13.88 | 14.29 | 14.30 | | 0.75 | 0.81 | 1.02 | 0.87 |
| 5 | *Nephelomys* | 13.14 | 13.00 | 13.78 | 10.34 | | 0.71 | 0.82 | 0.72 |
| 6 | *Oecomys* | 12.65 | 14.04 | 13.64 | 13.66 | 12.90 | | 0.91 | 0.76 |
| 7 | *Pattonimus* gen. nov. | 14.41 | 14.43 | 15.64 | 12.46 | 11.91 | 14.21 | | 1.00 |
| 8 | *Transandinomys* | 12.75 | 14.94 | 14.68 | 15.04 | 13.70 | 13.95 | 14.99 | |
| **Within the new genus** | | | | | | | | | |
| 1 | Reserva Drácula (sp. 1) | | 0.87 | 0.83 | | | | | |
| 2 | Reserva Rio Manduriacu (sp. 2) | 7.87 | | 0.96 | | | | | |
| 3 | Reserva Otonga (sp. 3) | 7.55 | 7.28 | | | | | | |

Note:
Uncorrected genetic distances in percentages (p-distances) between genera of clade B (sensu *Weksler, 2006*) of the Oryzomiynie tribe and among the three new species. The values on the diagonal represent the standard deviation.

monochrome-dark tails. The dorsal surface of the hindfoot is naked looking in the new genus, scarcely covered by short hairs in *Nephelomys* and *Tanyuromys* while it is densely covered by short hairs in *Mindomys*. Pes are relatively long (range: 35–36 mm) and narrow in the new genus, similar to *Nephelomys* (range: 30–42 mm) and *Tanyuromys* (range: 30–37 mm), but distinct from *Mindomys*, which exhibit very long (range: 38–42 mm) but much wider pes, configuring a shorter appearance (*Weksler, 2006*; *Percequillo, 2015*).

*Nephelomys* skulls are characterized by moderately deep and wide zygomatic notches, while these are noticeably shallower and narrower in *Mindomys* and in the new genus (Fig. 3). The new genus also exhibits a narrower and longer rostrum when compared to *Mindomys* and *Nephelomys*. The interorbital region is anteriorly convergent, with sharp supraorbital margins in the new genus, and hourglass-shaped, slightly convergent anteriorly or posteriorly with rounded or squared margins in *Nephelomys*, and slightly anteriorly convergent, with squared, beaded or slightly crested margins in *Mindomys*. In the new genus, the posterolateral palatal pits are single and small, while in *Mindomys* the pits are numerous and recessed in a shallow palatine depression. In *Nephelomys* the pits are also numerous, but variably positioned at the palate level, or from shallow to deeply excavated palatine depressions (that also vary form narrow and oblique to wide and round). The alisphenoid strut is present in all specimens of the new genus (Fig. 4), configuring separated buccinator-masticatory and ovale accessory foramen, but is variably present in species of *Nephelomys* (present in most individuals of *N. moerex*, and absent in most specimens of *N. devius*), and absent in specimens of *Mindomys* and *Tanyuromys*. The subsquamosal fenestra is small in the new genus, well-developed in *Nephelomys*, and absent in *Mindomys* and *Tanyuromys*. The squamosal ridge is absent in the new genus, present in *Nephelomys* and *Tanyuromys*, and barely present in *Mindomys* (Fig. 3).

**Table 2 Morphological comparisons.** Morphological comparisons of selected traits among *Pattonimus* gen. nov. and other related oryzomyines.

| | *Pattonimus* | *Nephelomys*[1] | *Mindomys*[2] | *Tanyuromys*[3] |
|---|---|---|---|---|
| Dorsal hindfeet condition | Naked-looking | Scarcely covered by short hairs | Densely covered by short hairs | Scarcely covered by short hairs |
| Rostrum | Moderate | Long | Moderate | Short |
| Zygomatic notch | Shallow | Well defined | Indistinct | Shallow |
| Lacrimal | Small | Medium | Medium | Small |
| Interorbit | Anteriorly convergent, with sharp margins | "hourglass," with rounded margins | Anteriorly convergent, with beaded margins | Anteriorly convergent, with sharp margins |
| Antorbital bridge | Broad | Narrow | Broad | Narrow |
| Molars relative size | Medium | Small | Large | Large |
| Incisive foramen relative size | Medium | Small | Medium | Medium |
| Incisive foramen maxillary septum | Narrow | Narrow | Narrow | Broad |
| Palate | Short | Long | Short | Short |
| Posterolateral palatal pits | Scarce | Numerous | Scarce | Scarce |
| Basioccipital | Long | Long | Short | Short |
| Zygomatic plate upper border | Not patent | Patent | Not patent | Not patent |
| Squamosal fenestra | Small | Well-developed | Absent | Absent |
| Lacerate foramen | Scarcely ossified | Scarcely ossified | Scarcely ossified | Well-ossified |
| Alisphenoid strut | Present | Unilaterally present | Absent | Absent |
| Squamosal ridge | Absent | Present | Barely present | Present |
| Sphenofrontal foramen | Covered by alar fissure | Not covered by alar fissure | Covered by alar fissure | Absent |
| Molar design | Incipiently laminate | Not laminated, bulbous | Not laminated, bulbous | Not laminated, bulbous |
| Enamel borders lophs and lophids | Straight | Straight | Straight | Crenulate |
| M1 procingulum | Compressed without flexus | Broad with flexus | Broad without flexus | Broad with flexus and fossete |
| M1–M2 anteroloph | Small or absent | Patent | Patent | Patent |
| M3 size relative M2 | M3 ≪ M2 | M3 < M2 | M3 < M2 | M3 < M2 |
| M3 shape | Subtriangular, compressed | Not compressed | Not compressed | Not compressed |
| M1 procingulum | Compressed, without flexid | Broad, with flexid | Broad, with fossetid | Broad, with fossetid |
| M1 anterior murid | Absent | Present | Present | Present |
| Mesolophids M1–M2 | Absent | Present | Present | Present |
| Angular process | Medium | Medium | Short and broad | Short and broad |
| Number of ribs | 12 | 12 | ? | 12 or 13 |

**Notes:**
[1] Character states are those of *Nephelomys albigularis*; other species currently classified under the genus may possess different attributes.
[2] Character states are those of *Mindomys hammondi*.
[3] Character states are those of *Tanyuromys thomasleei*; other species currently classified under the genus may possess different attributes.

The molar rows are medium sized with respect to the skull size in the new genus, being shorter in *Nephelomys*, and longer in *Mindomys* and *Tanyuromys*. The molar design is moderately laminated in the new genus (Fig. 5), but definitively not laminated in

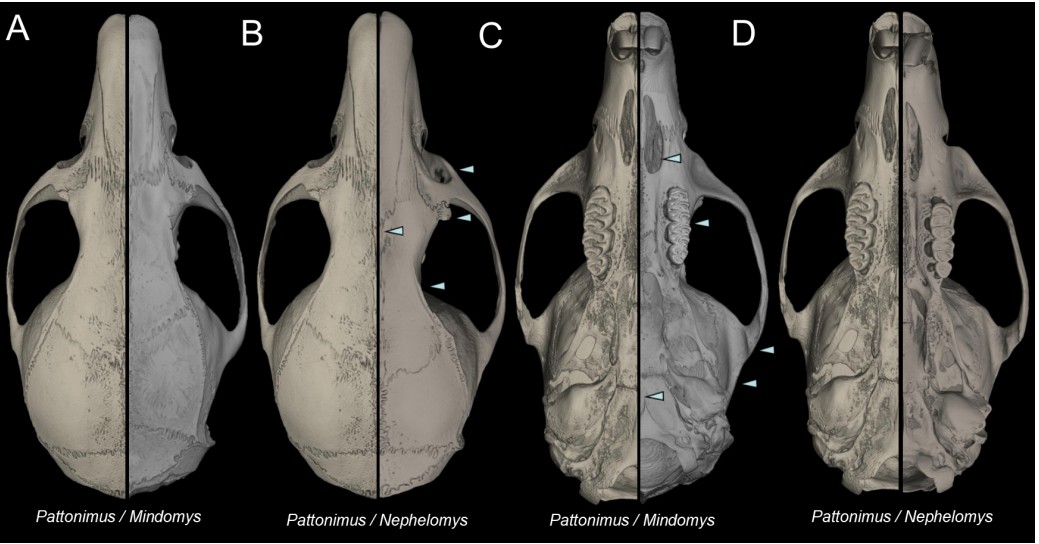

**Figure 3 Selected aspects of qualitative anatomy contrasted.** Selected aspects of qualitative anatomy contrasted in the crania of *Pattonimus* gen. nov. (left half, A–D; MECN 5928, holotype of *Pattonimus ecominga* sp. nov., genotype) vs *Mindomys hammondi* (right half, A and C; BM 13.10.24.58, holotype) and *Nephelomys auriventer* (right half, B and D; MECN 5812), scaled to the same length. The figure portrays contrasts between several characteristics highlighted by pointers.

*Nephelomys*, *Mindomys*, and *Tanyuromys*. The enamel borders of lophs and lophids in all molars are smooth in the new genus and in *Nephelomys* and *Mindomys*, while in *Tanyuromys* the borders are crenulate. The procingulum of M1 is compressed, without anteromedian flexus in the new genus (Fig. 5), broad with deep anteromedian flexus in *Nephelomys*, broad without flexus in *Mindomys*, and broad with flexus and anterior fossete in *Tanyuromys*. The anterolophs of M1–M2 are small or absent in the new genus and present in *Nephelomys*, *Mindomys*, and *Tanyuromys*. There is a perceptible variation in the size of M3 relative to the size of M2: in the new genus M3 < M2, while in *Nephelomys*, *Mindomys* and *Tanyuromys* M3 < M2. The mesolophs of M1–M2 are absent or poorly developed in the new genus (Fig. 5), but present and well developed in M1–M2 of *Nephelomys*, *Mindomys*, and *Tanyuromys*. The procingulum of m1 is compressed and lacks the anteromedian flexid in the new genus, but is broad with flexid in *Nephelomys*, and broad with an anterior fossetid in *Mindomys* and *Tanyuromys*. The anterior murid of m1 is absent in the new genus, but present in *Nephelomys*, *Mindomys* and *Tanyuromys*. The m3 is subtriangular and compressed in the new genus, while not compressed in *Nephelomys*, *Mindomys* and *Tanyuromys*. The accessory root of M1 is present in the new genus and in *Tanyuromys* (Supplemental Information S7), but absent in *Nephelomys* and *Mindomys*. The accessory root of m1, and two accessory roots of m2–m3 are present in *Tanyuromys*, while absent in the new genus, and in *Nephelomys* and *Mindomys* (Supplemental Information S7).

**Geographic variation:** Studied samples of the new genus came from different montane forest blocks distributed in the Pacific slope of the Andean Cordillera Occidental (Ecuador and Colombia). As this humid and cold forest band is transversally interrupted

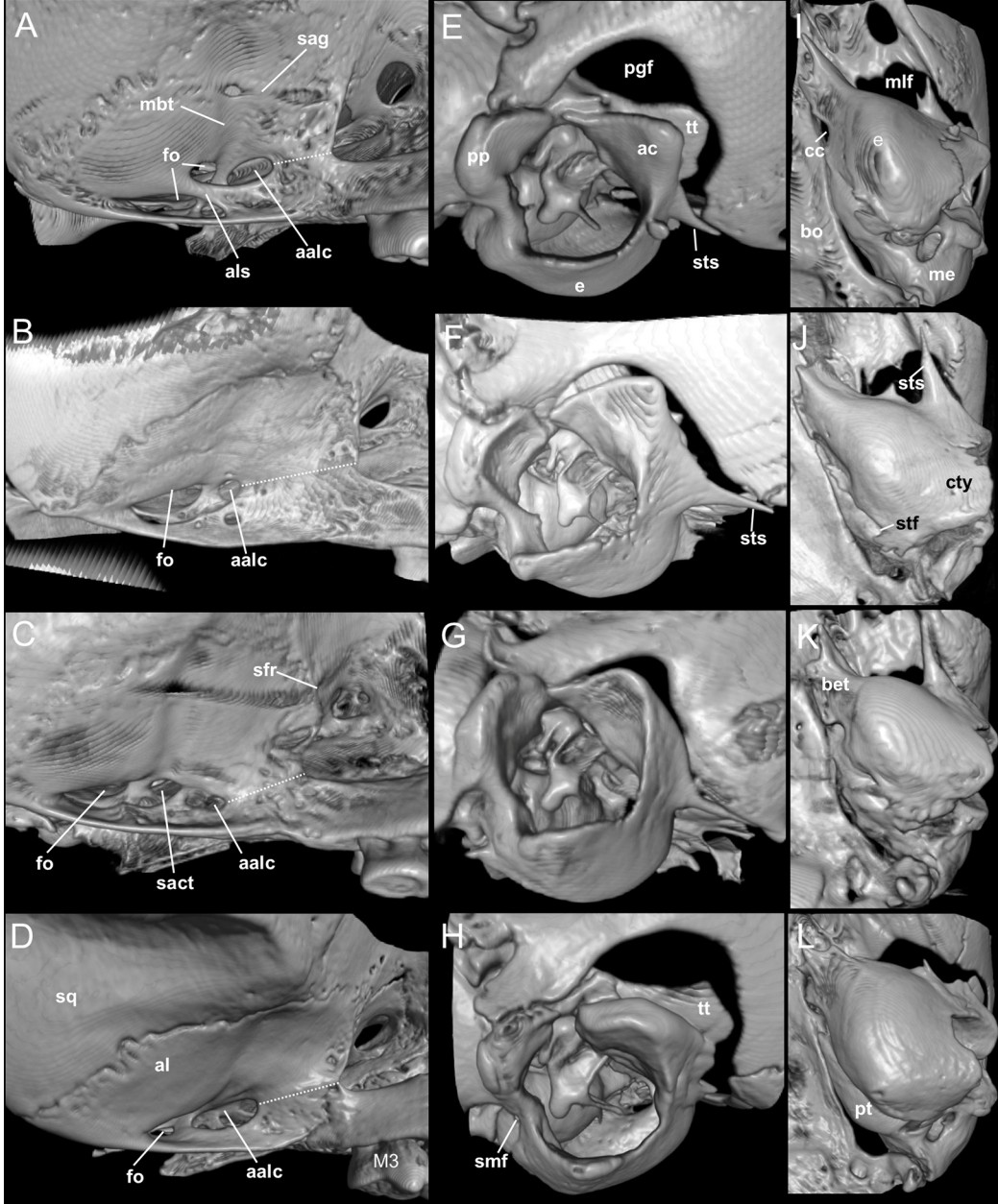

**Figure 4 Comparison of selected anatomical regions of the cranium.** Comparison of selected anatomical regions of the cranium of *Pattonimus* gen. nov. (A, E and I; MECN 5928, holotype of *Pattonimus ecominga* sp. nov., genotype), *Mindomys hammondi* (B, F and J; BMNH 13.10.24.58, holotype), *Nephelomys auriventer* (C, G and K; MECN 5812) and *Tanyuromys thomasleei* (D, H and L; MECN 3407). Right squamosal-alisphenoid region in lateral view (left), right auditory region in lateral view (middle) and right auditory capsule in ventral view (right). Abbreviations: aalc, anterior opening of alisphenoid canal; ac, anterior crus of ectotympanic; al, alisphenoid; als, alisphenoid strut; bet, bony eustachian tube; bo, basioccipital; cc, carotid canal; cty, crista tympanica; e, ectotympanic; fo, foramen ovale; mbt, trough for masticatory-buccinator nerve; me, mastoid exposure; mlf, middle lacerate foramen; pgf, postglenoid foramen; pp, paroccipital process of petrosal; pt, petrosal; sact, tunnel for secondary arterial connection between internal carotid and orbital-maxillary circulation; sag, squamosal alisphenoid groove; sfr, sphenofrontal foramen; smf, stylomastoid foramen; sq, squamosal; stf, stapedial foramen; sts, stapedial process of bulla (rostral process of malleus?); tt, tegmen tympani.

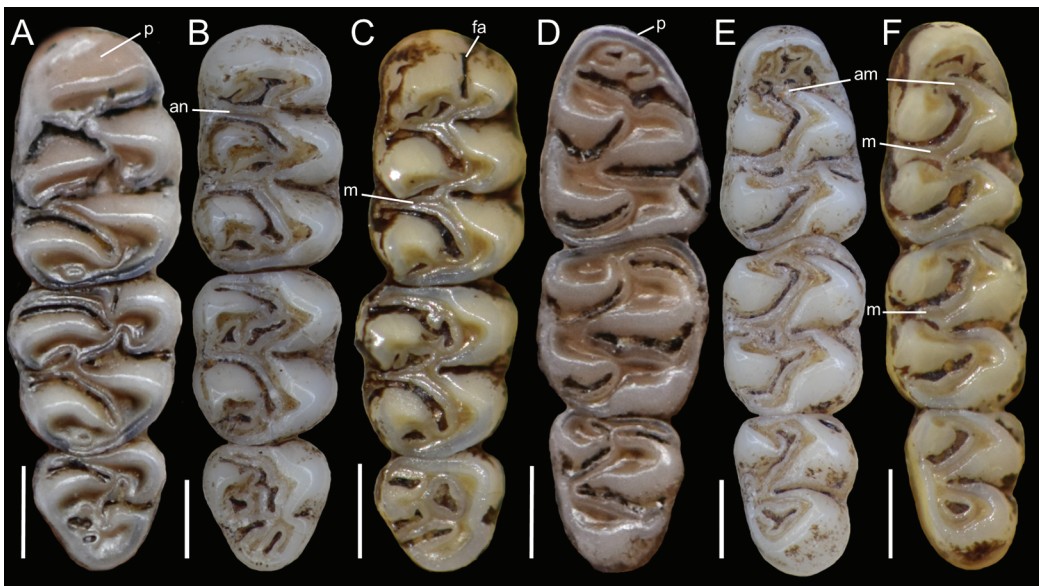

**Figure 5 Lower right toothrows in occlusal view.** (A–C), Upper and (D–F), lower right toothrows in occlusal view of *Pattonimus* gen. nov. (A and D; MECN 5928, holotype of *Pattonimus ecominga* sp. nov., genotype), *Mindomys hammondi* (B and E; BMNH 13.10.24.58, holotype) and *Nephelomys albigularis* (C, F; MECN 583). Abbreviations: an, anteroloph; am, anterior murid; fa, anteromedian flexus; m, mesoloph/id; p, procingulum. Scale = 1 mm.

by several important east-to-west river canyons (from north to south, Güiza, Mira, Guayllabamba, and Toachi; Fig. 6), we focused on the detection of potential morphological differences among examined populations, under the assumption that habitat discontinuities promote allopatric speciation. This inspection was directed to the animals collected in Reserva Drácula and Reserva Río Manduriacu, the two largest available collections. The collection from Reserva Otonga is composed of a single young specimen with the third molars not fully erupted and was thus discarded from the analysis. The northernmost samples of the new genus, two individuals from Colombia, were also discarded, because we were unable to review the voucher material.

Specimens from Drácula (*n* = 12) and Río Manduriacu reserves (*n* = 4) are externally very similar, although the fur of the latter is less dorsoventrally countershaded because of the grayer bellies. These chromatic differences are also displayed in the tails, which are darker above and below in animals from Río Manduriacu. In contrast, several cranial and dental traits exhibit fixed differences between both samples. Drácula specimens are characterized by a broad dorsal expression of the antorbital bridge, an alar fissure typically without a basal notch, and a small but constant participation of the parietals in the lateral wall of the cranium. In contrast, the antorbital bridge from specimens of Río Manduriacu is dorsally narrow, the alar fissure has a marked basal notch, and the lateral expansions of the parietals are absent. Conspicuous differences between both samples are even better expressed in the dentition. The enamel of the upper incisors are cream or white-colored in animals from Drácula, while those of specimens from Río Manduriacu are bright orange-colored. In addition, probably due to a slight difference in hypsodonty,

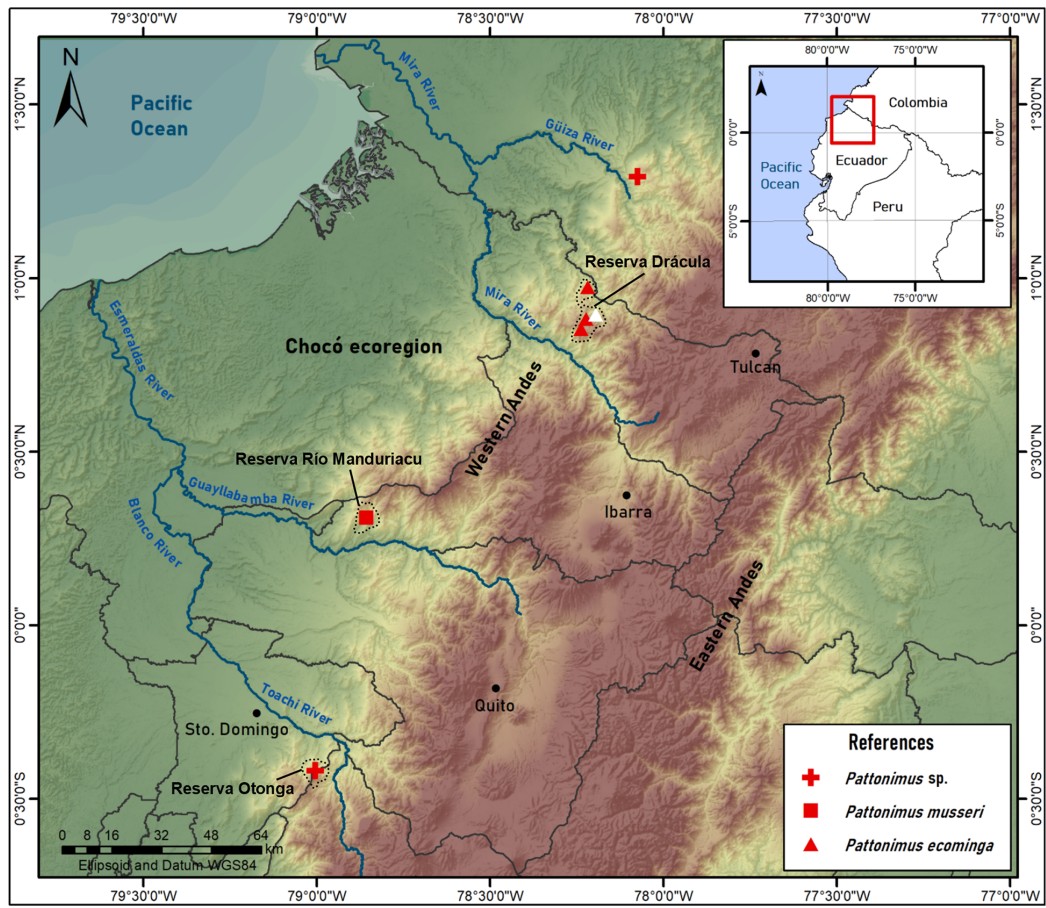

**Figure 6 *Pattonimus* gen. nov., geographic distribution in Ecuador and Colombia.** *Pattonimus* gen. nov. (Oryzomyini, Sigmodontinae), geographic distribution in Ecuador and Colombia. The white triangle represent the type locality.               

occlusal design in Drácula specimens is moderately complex showing incipient anterolophs and well developed mesolophs in both M1 and M2. Conversely, Río Manduriacu specimens typically lack both structures.

Since molars accounted for several important morphological differences between both populations, we calculated Mahalanobis distances on molar individual measurements and performed a cluster analysis. The obtained result grouped the examined animals separately according to geographic provenance, reinforcing the taxonomic hypothesis that we are dealing with two differentiable entities of specific rank (Supplemental Information S8). To further explore metrical differences, we used principal component analysis to summarize patterns of multivariate craniodental variation. The first two principal components accounted for about 54% of the total variance (Supplemental Information S9). Projected specimen scores indicated a poor sample separation of the first two components, the coefficients of which suggested that Río Manduriacu specimens differ from Drácula specimens by their slightly longer ears and comparatively shorter molars. To strengthen the morphometric analysis, we ran a PCA based exclusively on measurements showing significant univariate differences ($p < 0.05$) between both samples;

these were CIL, BBP, length of M2, and length of m1. As expected, the variance explained by the first two principal components increased to about 75% and the distribution of the specimens in the multivariate space showed that the animals from Río Manduriacu are smaller than those of Drácula (Supplemental Information S9). When individuals from southern Colombia were included in a PCA performed with a matrix composed of those variables with significant differences ($p < 0.05$; CIL, LD, BIF, BR, LN, ZB, and BZP), similar results were obtained (Supplemental Information S9). In this instance, Colombian animals were grouped separately, highlighting their larger size, a fact that can be assessed from a direct inspection of Table 3. Although two individuals obtained the craniodental measurements (JB measured Ecuadorian animals while ARP measured Colombian ones), differences were well beyond expected differences due to methodological bias. Finally, a PCA (Fig. 7) was conducted on a matrix exclusively composed of all measured craniodental variables (i.e., excluding external measurements in order to avoid the potential negative effect of mixing larger and smaller dimensions). Although the variance explained by the first two principal components reached almost 85%, one of the two specimens from Rio Manduriacu was placed among those of Drácula Reserve (Supplemental Information S9).

At the phylogenetic level, specimens from Rio Manduriacu and the Drácula Reserve were allocated to different monophyletic clades (Supplemental Information S5 and S6), and in turn these were recovered as sister clades (Figs. 1 and 2). These clades show a considerable genetic divergence (7.90% higher), allowing these populations to be considered different taxonomic entities.

Summarizing, we interpret these overall results to indicate the presence of two species of the new genus under discussion, one in the forest of Reserva Drácula and the other in the forest of Reserva Río Manduriacu. Fortunately, morphological qualitative and quantitative evidence is in accordance with the clear separation of these samples on molecular grounds. Regarding the Colombian specimens, they seem to be metrically larger than the Ecuadorian specimens, but further studies are needed to establish their taxonomy.

The data presented indicate that the new rice rats discussed here are representatives of a new genus of Oryzomyini. In addition, we advance evidence of at least two, and possibly three, species within this new genus. We provide below a definition of the genus, followed by a description and a discussion of its relationships, and morphological descriptions of two of the recognized species.

## SYSTEMATIC ACCOUNTS

Family Cricetidae Fischer, 1817
Subfamily Sigmodontinae Wagner, 1843
Tribe Oryzomyini Vorontsov, 1959
**Pattonimus** gen. nov.
urn:lsid:zoobank.org:act: 83926983-C0A8-4337-B5F9-81B01CF7B487
Patton's montane rat, Rata montana de Patton
*Sigmodontomys*: *Cadena, Anderson & Rivas-Pava* (*1998*: 11), part, not *Sigmodontomys* Allen, 1897

**Table 3 Individual external craniodental measurements (in mm).** Individual external craniodental measurements (in mm, except body mass) of the paratypes of *Pattonimus ecominga* sp. nov. and *Pattonimus musseri* sp. nov. and the material referred as *Pattonimus* sp. (Oryzomyini, Sigmodontinae).

| | *P. ecominga* sp. nov. | | | | | | | | | | *P. musseri* sp. nov. | | | *Pattonimus* sp. | |
|---|---|---|---|---|---|---|---|---|---|---|---|---|---|---|---|
| Collection | MECN | MECN | MECN | MECN | MECN | MECN | MECN | MECN | MECN | MECN | MEPN | MEPN | MEPN | ICN | ICN |
| Number | 5927 | 6017 | 6019 | 6020 | 6025 | 6040 | 6041 | 6042 | 6043 | 6173 | 12586 | 12587 | 12593 | 13663 | 21487 |
| Sex | F | M | M | F | M | F | M | M | F | F | F | M | M | M | F |
| Age | 3 | 1 | 3 | 3 | 4 | 3 | 3 | 3 | 3 | 3 | 3 | 5 | 0 | 3 | 4 |
| HB | 138.00 | 103.00 | 120.00 | 110.00 | 130.00 | 120 | 117 | 118 | 120 | 115 | 115 | 130 | 95 | 136.00 | 140.00 |
| TL | 188.00 | 135.00 | 170.00 | 171.00 | 180 | 171 | 155 | 160 | 155 | 156 | 139 | 190 | 120 | 184.00 | 180.00 |
| HF | 33.00 | 33.00 | 33.00 | 31.00 | 34 | 35 | 35 | 33 | 33 | 34 | 32 | 35 | 28 | 36.00 | 35.20 |
| E | 19.00 | 15.00 | 18.00 | 14.00 | 16 | 16 | 16 | 16 | 15 | 16 | 17 | 19 | 17 | 20.00 | 15.50 |
| LMV | 50.23 | 53.00 | 53.00 | 56.00 | 60 | 54 | 52.1 | 55 | 50.94 | 51 | 51.22 | 60.3 | 33.4 | – | – |
| LSV | 30.00 | 25.00 | 29.00 | 27.00 | 31 | 28 | 19.64 | 26 | 27.65 | 25 | 17.59 | 29.05 | 21.47 | – | – |
| LGV | 25.00 | 18.00 | 19.00 | 23.00 | 21 | 22 | – | 18 | 20.7 | 20 | 19.45 | 16.58 | 17.23 | – | – |
| W | 64.00 | 30.00 | 57.00 | 60.00 | 83 | 76 | 64 | 55 | 54 | 47 | 44.5 | 110 | 25 | 72.00 | 68.00 |
| ONL | 33.67 | – | 32.29 | 31.16 | 34.6 | 32.94 | – | 32.27 | 31.43 | 30.38 | 29.61 | 36.73 | 26.21 | – | – |
| CIL | 31.22 | 24.66 | 29.98 | 29.27 | 32.1 | 30.48 | 29.44 | 30.03 | 28.73 | 28.36 | 26.85 | 33.75 | 24.53 | 31.63 | 31.33 |
| LD | 8.43 | 6.48 | 8.40 | 8.04 | 8.94 | 8.02 | 7.92 | 8.20 | 8.14 | 7.50 | 7.18 | 9.67 | 6.35 | 9.37 | 8.88 |
| LUM | 5.71 | 5.35 | 5.50 | 5.56 | 5.41 | 5.61 | 5.72 | 5.60 | 5.50 | 5.53 | 5.56 | 5.83 | – | 5.50 | 5.60 |
| LIF | 4.06 | 3.43 | 4.33 | 4.3 | 4.96 | 4.45 | 4.71 | 4.53 | 4.46 | 4.03 | 3.5 | 4.47 | 3.48 | 4.94 | 5.43 |
| BIF | 1.75 | 1.66 | 1.73 | 1.84 | 1.7 | 1.71 | 1.93 | 1.74 | 1.92 | 1.56 | 1.47 | 1.84 | 1.58 | 1.96 | 2.25 |
| BM1 | 1.79 | 1.67 | 1.79 | 1.75 | 1.77 | 1.74 | 1.8 | 1.73 | 1.68 | 1.73 | 1.80 | 1.80 | 1.70 | 1.74 | 1.77 |
| BR | 6.70 | 5.13 | 6.22 | 6.01 | 6.27 | 5.98 | 5.89 | 5.90 | 5.87 | 5.86 | 5.30 | 6.80 | 5.22 | 5.70 | 5.47 |
| LN | 13.02 | – | 12.62 | 12.16 | 13.08 | 12.06 | 11.74 | 11.45 | 11.55 | 11.07 | 10.63 | 13.32 | 9.24 | 13.2 | 13.74 |
| LPB | 7.53 | 6.09 | 7.26 | 6.98 | 7.75 | 7.15 | 6.60 | 6.92 | 6.27 | 6.77 | 6.92 | 8.05 | 5.49 | 7.36 | 7.02 |
| BBP | 2.71 | 2.22 | 2.78 | 2.64 | 2.22 | 2.63 | 2.60 | 2.74 | 2.73 | 2.54 | 2.21 | 2.63 | 2.08 | – | – |
| LIB | 5.79 | 5.58 | 6.04 | 5.76 | 5.72 | 5.9 | 5.83 | 5.62 | 5.77 | 5.86 | 5.56 | 5.73 | 5.56 | 6.09 | 5.76 |
| ZB | 16.76 | 14.40 | 16.55 | 16.22 | 17.61 | 16.57 | 16.40 | 16.38 | 15.80 | 15.96 | 14.8 | 17.1 | 14.53 | 17.67 | 17.37 |
| BZP | 3.41 | 2.97 | 3.62 | 3.43 | 4.02 | 3.45 | 3.33 | 3.48 | 3.28 | 3.46 | 3.35 | 4.20 | 2.62 | 4.01 | 3.97 |
| LB | 13.33 | 12.40 | 13.14 | 12.57 | 12.95 | 12.68 | 12.85 | 13.08 | 12.89 | 12.6 | 12.22 | 13.77 | 12.12 | – | – |
| OFL | 10.34 | 8.86 | 10.08 | 9.90 | 11.1 | 10.54 | 10.39 | 10.53 | 9.94 | 9.69 | 9.57 | 11.20 | 8.76 | 11.03 | 10.93 |
| BB | 3.80 | – | 3.83 | – | 3.82 | 3.82 | 3.75 | 3.82 | 3.70 | 3.88 | 3.65 | 3.81 | 3.53 | – | – |
| LM | 17.91 | 15.30 | 17.54 | 16.63 | 17.90 | 17.38 | 16.68 | 16.81 | 16.73 | 16.73 | 15.16 | 18.98 | 14.82 | – | – |
| LLM | 5.51 | 5.46 | 5.50 | 5.68 | 5.46 | 5.52 | 5.63 | 5.53 | 5.49 | 5.61 | 5.49 | 5.64 | – | – | – |
| LLD | 3.76 | 3.95 | 4.28 | 3.77 | 4.16 | 3.72 | 3.8 | 4.1 | 3.98 | 3.98 | 3.72 | 5.13 | 3.98 | | |

*Mindomys*: *Pinto et al. (2018*: figs. 2, 5, and Appendix); part, not *Mindomys Weksler, Percequillo & Voss, 2006*.

**Type species:** *Pattonimus ecominga* sp. nov.

**Diagnosis:** A medium-sized (adult combined head and body length ~ 130 mm; body mass ~ 60 grams; condyle-incisive length ~ 30 mm; coronal maxillary toothrow length ~ 5.6 mm)

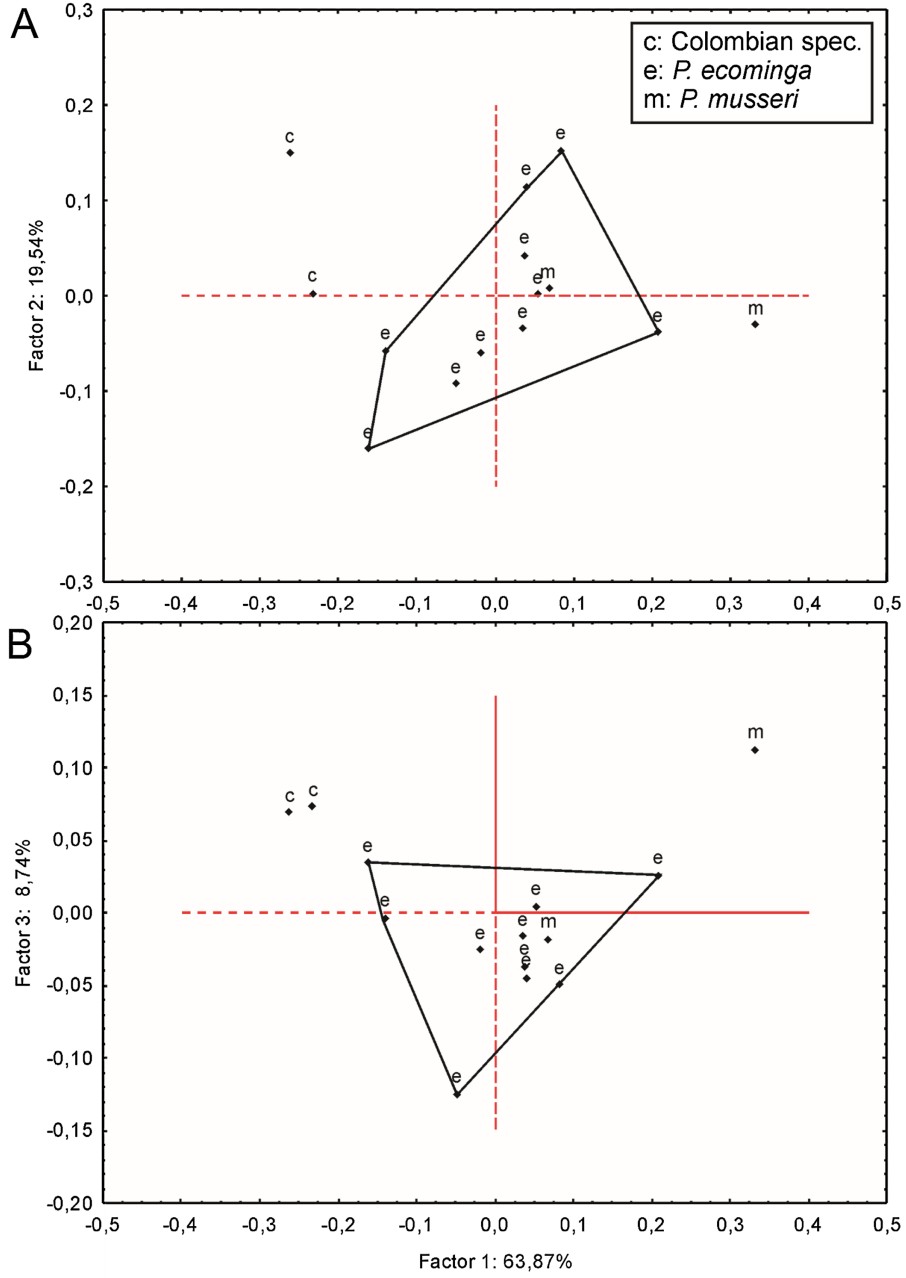

**Figure 7 Principal component analysis.** Scatter plot (A and B) of principal component analysis (PCA) of *Pattonimus* gen. nov.

member of the tribe Oryzomyini characterized by the following combination of characters: body pelage short and close, reddish brown dorsally (dorsal hairs with agouti banding pattern) with a subtle darker middorsal stripe, ventral hairs plumbeous washed with yellowish tones, weak countershading; mystacial vibrissae abundant and longer than ears when laid backwards; ears rounded, haired and small; tail longer than combined length of head-and-body (130%) and naked in appearance, unicolored; 8 mammae; cranium with moderately long (~35% of the occipitonasal length) and wide rostrum,

shallow zygomatic notches, interorbital region with sharp frontals anteriorly convergent, zygomatic plate broad and excavated; carotid arterial pattern primitive, alisphenoid strut present, short and broad hamular process of squamosal, incisive foramen short, teardrop-shaped, well anterior to the M1s anterior faces, palate narrow and short, posterior palatal foramen inconspicuous, broad mesopterygoid fossa, broader than parapterygoid plates; maxillary toothrows slightly divergent backwards; molars large, with tendency towards lamination, moderate hypsodonty and simplification; procingulum of M1 anteriorly-posteriorly compressed, mesolophs/ids absent to moderately developed, hypocone connected to paracone, procingulum of m1 undivided, typically with a central fossetid, anterior murid typically absent (protoflexid confluent with metaflexid), anterolabial cingula strongly developed in all upper and lower molars, m3 anteriorly-posteriorly compressed; first rib with dual articulation (seventh cervical and first thoracic vertebrae), 12 ribs; second thoracic vertebra with elongated neural spine; 19 thoracicolumbar vertebrae; 4 sacrals, 34–36 caudals, the first four with hemal arches; stomach unilocular-hemiglandular, with glandular epithelium extended to the corpus; gall bladder absent.

**Morphological description:** Adult body fur fine and short (dorsal hairs averages 7–8 mm), moderately soft, but not woolly; black guard hairs extend slightly beyond the body fur, not much longer than the regular coat except on the rump; upperparts and underparts are sharply delineated; dorsal fur reddish brown with a subtle darker middorsal stripe; individual overhairs exhibit an agouti banding pattern (basal three-fourths plumbeous, followed by an ochraceous-reddish band, then a blackish tip), usually darker on rump; flanks tending to more reddish; ventral pelage paler agouti, sometimes grayish; head with marked brown-darker fur reaching the rhinarium; whitish gular patch; eyes small. Mystacial, superciliary, genal, submental, interramal, and carpal vibrissae present; mystacial vibrissae abundant (about 20 per side) and long, some extending posteriorly beyond caudal margins of pinnae when laid back against cheeks; ears large and clearly visible above fur of head, moderately clothed with soft reddish hairs on the basal third externally, the rest nearly naked (sparsely covered with very short reddish hairs) on both surfaces; helix and antitragus poorly developed (Fig. 8E). Upper lips densely covered with whitish hispid hairs; rhinarium with well-developed nasal pads; philtrum present (Fig. 8F). The tops of the fore and hindfeet are almost naked, poorly covered with scarce and fine whitish hairs; digits naked; except plantar digit 1 (hallux), the end of each one bears a few silvery hairs which slightly surpass the tip of the claw; manus ventral surface naked, finely scutellate and sometimes dark pigmented, with five fleshy plantar tubercles (Fig. 8B); claws short, unusually recurved, basally opened, except the pollex which bears a nail; pes long and narrow, with outer digits (1 and 5) much shorter than middle three (claw of d1 extending to middle of first phalange of d2, claw of d5 extending just beyond first interphalangeal joint of d4); plantar surface naked, dark pigmented, with finely squamae (scale-like tubercles) and complete pad dotation (2 metatarsal and 4 interdigital tubercles; Fig. 8D). Tail longer than combined length of head-and-body (130%), apparently naked but with 3 fine and rigid very short hairs per scale (Fig. 8I), and

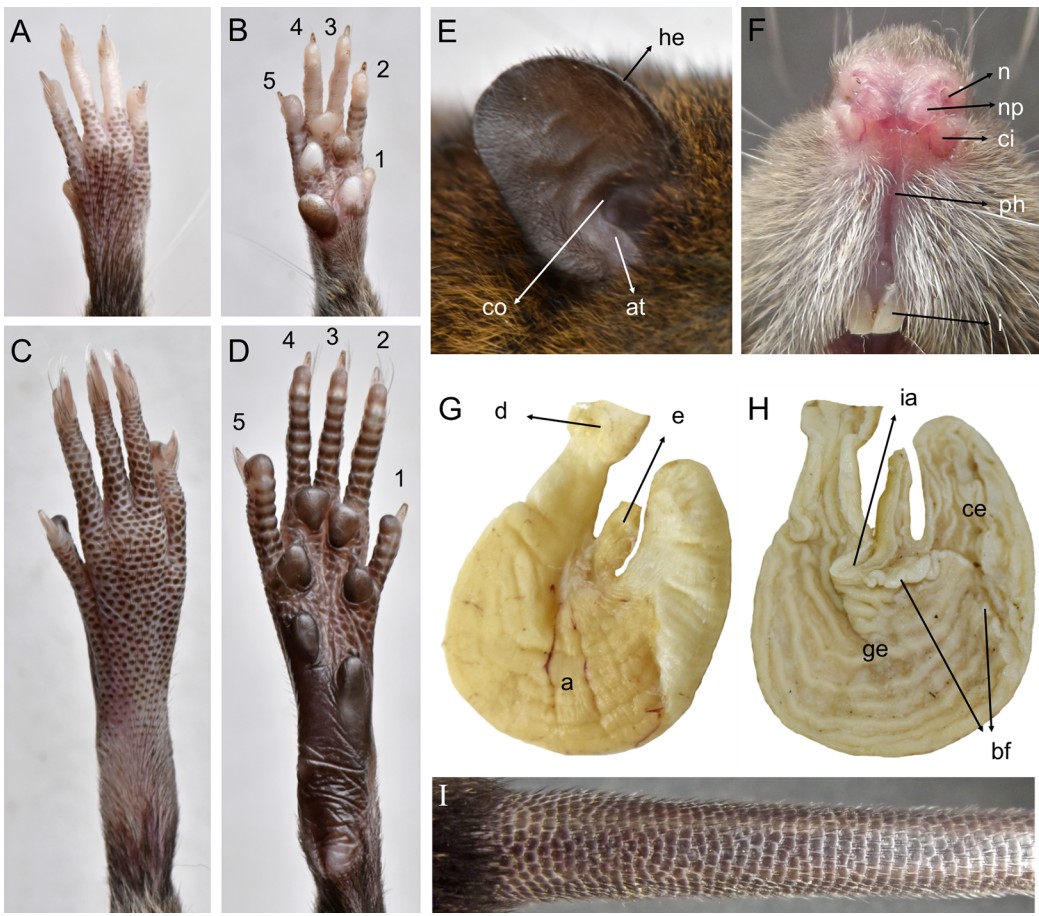

**Figure 8 *Pattonimus* gen. nov. (Oryzomyini, Sigmodontinae), selected features of external and internal anatomy.** *Pattonimus* gen. nov. (Oryzomyini, Sigmodontinae), selected features of external and internal anatomy (based on MECN 5928, holotype of *P. ecominga* sp. nov., genotype): (A) dorsal and (B) plantar surface of the right manus; (C) dorsal and (D) plantar surface of the right pes; (E) right ear, internal view; (F) rhinarium, ventral view; (G and H) stomach, mid-dorsal portions in external and internal view, respectively; (I) tail, anterior portion in dorsal view. Abbreviations: 1–5, digits; a, antrum; at, antitragus; bf, bordering fold; ce, cornified epithelium; ci, crus inferius of the narial pad; co, concha; d, duodenum; e, esophagus; ge, glandular epithelium; he, helix; i, incisive; ia, incisura angularis; n, nostrils; np, nasal pads; ph, philtrum.

unicolored (dark above and below), slightly paler below. Mammae 8 in inguinal, abdominal, postaxial, and pectoral pairs.

Skull with moderately long (about 35% of ONL) and wide rostrum, the greater width results from the relatively inflated nasolacrimal capsules and the broad premaxillaries; rostral sides taper gradually forward from nasolacrimal capsules, but premaxillary bones can be seen to extend for almost their entire length along the nasal margins except its distal portion hidden beneath nasals; nasals gradually diverging forward with distal end moderately upturned; shallow but distinct zygomatic notches (Fig. 3); notable internal bony development in the respiratory and olfactory sagittal plane: two frontoturbinals, one interturbinal and three ethmoturbinals present (Supplemental Information 10); interorbit wide, anteriorly convergent with sharp but not beaded supraorbital margins, extending

posteriorly, concealing external sutures of parietals and interparietal, and imparting a "tennis racket" appearance to braincase in dorsal view; fronto-parietal suture ranging from V-shaped to U-shaped; braincase moderately inflated and elongated, with marked temporal crests; cranial roof dorsal profile flat from nasals to the half of parietals to slope sharply downward toward the occiput; foramen magnum is oriented posteroventrad and the occipital condyles are inconspicuous viewed from above; interparietal well developed, covering almost the entire rear portion of the braincase, flanked by exoccipitals. Premaxillae slightly shorter than nasals, not produced anteriorly beyond incisors to form a rostral tube; gnathic process small but distinct; zygomatic plate broad and excavated, its anterior edge slightly sloping forward, with an angular anterodorsal contour and a thick antorbital bridge; antorbital foramen basally narrowed; zygomatic arches sturdy and robust with jugals spanning a short segment of each mid-arch but distinctly separating zygomatic processes of the maxillary and squamosal bones; maxillary extension of the zygomatic arch with a typically patent projection in its lower border; zygomatic arches with ventralmost projection above the floor of the orbit; squamosal-alisphenoid groove poorly visible through the translucent braincase, usually without a perforation where it crosses the depression for the masticatory nerve, leading to a small sphenofrontal foramen sometimes hidden by the alar fissure; large stapedial foramen and carotid canal but barely expressed petrotympanic fissure; primitive cephalic arterial supply (pattern 1 of *Voss, 1988*); alisphenoid strut consistently present, separating buccinator-masticatory foramen and foramen ovale (Fig. 4); postglenoid foramen narrow separated from an also narrow subsquamosal fenestra by short and broad hamular process of squamosal; well-developed tegmen tympani mostly covering subsquamosal fenestra and contacting squamosal border but neither overlapping nor involving a distinct posterior suspensory squamosal process; squamosal root of zygomatic arch produced backwards as a short ridge well above the hamular process; small lateral expressions of parietals barely present; bullae small; pars flaccida of tympanic membrane present, large; orbicular apophysis of malleus well developed. Hill foramen moderately large; incisive foramina short, teardrop-shaped, averaging about 50% of diastemal length, well anterior to the M1s anterior faces; capsular process of premaxillary well developed and covering 2/3 of incisive foramina; palate narrow and short, almost uncomplicated (shallow lateral grooves), with the anterior border of the mesopterygoid fossa even with the plane defined by M3s posterior faces; posterior palatal foramen inconspicuous; small posterolateral pits usually paired and located side by side with the anterior part of the mesopterygoid fossa, never recessed in a common fossa; broad mesopterygoid fossa, broader than parapterygoid plates, with anterior margin U-shaped; bony roof of fossa complete; squared and short hamular processes of pterygoid sometimes contacting spinous processes of the bony Eustachian tubes; periotic well exposed.

Mandible moderately elongated, robust, with well-developed falciform coronoid process with its tip slightly surpassing the condyle level; mental foramen laterally placed; incisor case broad; inferior masseteric ridge well-marked, while superior masseteric ridge short and both conform an oblique and short common masseteric crest; condyle broad with well-developed pre- and postcondylid processes; lower incisor alveolus without distinct
capsular process on lateral mandibular surface; lunar notch poorly excavated; angular process short and broad.

Upper incisors ungrooved, opisthodont, narrow but deep, with yellow-orange (*P. musseri* sp. nov.) to cream (*P. ecominga* sp. nov.) enamel bands and straight dentine fissure. Maxillary molar rows slightly divergent backwards; upper molars large, with tendency towards lamination (labial and lingual reentrant folds long and interpenetrating) and moderate hypsodonty (Fig. 5); coronal surfaces slightly crested in young and subadults, tending planar in adults and old individuals; M1 > M2 ≫ M3 in length; main cusps slightly alternated and sloping backwards when viewed from side; M1 subrectangular in outline with procingulum not divided into labial and lingual conules, anteriorly-posteriorly compressed, without anteromedian flexus; anterior face rimmed by conspicuous enamel shelf; protocone isolated, connected to paracone through a minute enamel bridge; anteroloph barely present; mesoloph typically absent (*P. musseri* sp. nov.) to present (*P. ecominga* sp. nov.), and if present fused to minute mesostyles in adult individuals (in both species); posteroloph usually present as a small fossete; M2 squared in outline but posteriorly compressed with a procingulum limited to a labial loph; mesoloph, mesostyle, and posteroloph showing the same condition as in M1; M3 subtriangular in outline with an inconspicuous hypoflexus and a compressed posterior lobe. M1 four-rooted (with one accessory labial root but without external expression); M2 and M3 three-rooted.

Mandibular molars with main cusps alternated and sloping backwards when viewed from side. First mandibular molar (m1) with procingulum undivided, anteriorly-posteriorly moderately compressed, typically showing a large central fossetid of uncertain homology (probably formed from the fusion of two fossetids) and a well-developed labial cingulum fused to a protolophid which rarely closes the protoflexid; anterior murid barely present (protoflexid confluent with metaflexid). Procingulum of m1 not divided into labial and lingual conulids; metaflexid fused with the protoflexid; metaconid connected to protoconid through a narrow bridge; anterolophid indistinct; mesolophid absent in m1 and m2; m2 squared in outline; m3 triangular in outline with a deep hypoflexid and a compressed posterior lobe. All mandibular molars two-rooted.

Tuberculum of first rib articulates with transverse processes of seventh cervical and first thoracic vertebrae; second thoracic vertebra with differentially elongated neural spine; thoracicolumbar vertebrae 19, the 17th with moderately developed anapophyses; sacrals 4; caudals 34–36, with complete hemal arches in the first four; ribs 12; entepicondylar foramen of humerus absent; supratrochlear foramen of humerus present.

Gross stomach configuration (in three dissected specimens of *P. ecominga* sp. nov.) unilocular-hemiglandular (sensu *Carleton, 1973*), with a shallow but marked incisura angularis and with the limit (bordering fold) between internal epithelia crossing the organ clearly to left of the esophageal opening; therefore, the glandular lining is extended to corpus and has a folded internal surface (Figs. 8G and 8H). Gall bladder absent (according to three dissected specimens of *P. ecominga* sp. nov. and of one *P. musseri* sp. nov.).

Phallic, male reproductive characters, and karyotype undetermined.

**Contents**: Two species are described here as *Pattonimus ecominga* sp. nov., and *Pattonimus musseri* sp. nov.; one or possibly two additional species are presumably known.

**Geographic distribution**: Known from the western Andean cordillera of Colombia, Department of Nariño, and Ecuador, provinces of Carchi and Cotopaxi (Fig. 6), at elevations from ca. 1,200 to 2,350 m.

**Etymology**: The generic name (a noun in the nominative singular) is derived from the surname Patton and the Latin noun *mus* (= mouse, rat). This name honours the figure and legacy of James L. Patton, Emeritus Curator of Mammals and Professor of Integrative Biology, at the Museum of Vertebrate Zoology, University of California, in Berkeley, USA. James Patton inspired generations of mammalogists, through his adventurous field-trips and not so memorable shipwrecks, outstanding scientific contributions and supervision and mentoring of numerous students around the world (see *Patton, 2005*; *Rodríguez-Robles & Greene, 2005*).

*Pattonimus ecominga* sp. nov.
urn:lsid:zoobank.org:act:15A88558-F671-46C8-8826-D0E3962F620C
Ecominga montane rat, Rata montana de Ecominga

**Holotype:** MECN 5928 (field number JBM 2218), an adult male specimen preserved as a skull, partial postcranial skeleton and skin in regular condition; collected by Jenny Curay, Rocío Vargas, Camila Bravo, and Jorge Brito on 14 April 2019.

**Paratopotypes:** MECN 5927 (JBM 2223), an adult female, and MECN 6034 (JBM 2229), an adult male, both preserved as skull, partial postcranial skeleton and skin in regular condition; collected by J. Curay, R. Vargas, C. Bravo, and J. Brito between 15 and 17 April 2019.

**Other paratypes:** MECN 6017 (JBM 1936), a young female preserved as skull, partial postcranial (boneless autopodium) and skin in regular condition; collected in Pailón Alto (0.97415°N, 78.2176°W, 1,630 m) by J. Brito, J. Curay, and R. Vargas on 7 November 2017. MECN 5293 (JBM 1456), an adult male; MECN 5297 (JBM 1460), an adult male; MECN 5298 (JBM 1461), a young male; MECN 5304 (JBM 1467), a young male; MECN 5308 (JBM 1471), an adult female; MECN 5309 (JBM 1472), a young male; MECN 5310 (JBM 1473), an adult female; MECN 5325 (JBM 1488), an adult male; MECN 5326 (JBM 1489), an adult male; and MECN 5382 (JBM 1665), a young male. All these specimens were preserved as crushed skulls, crania and mandibles, partially covered by dry tissues, with carcass and viscera in ethanol. All of them were collected in Gualpi, Km. 18 of the Gualpi road (0.853841°N, 78.237600°W, 2,350 m) by J. Brito, J. Robayo, L. Recalde, T. Recalde and C. Reyes on 27 September 2016. MECN 6019 (JBM 2048), an adult male; MECN 6020 (JBM 2063), an adult male; MECN 6025 (JBM 2064), an adult male; MECN 6040 (JBM 2051), an adult female; MECN 6041 (JBM 2052), an adult female; MECN 6042 (JBM 2056), an adult male; and MECN 6043 (JBM 2057), an adult female. All these specimens were preserved as skull, partial postcranial skeleton

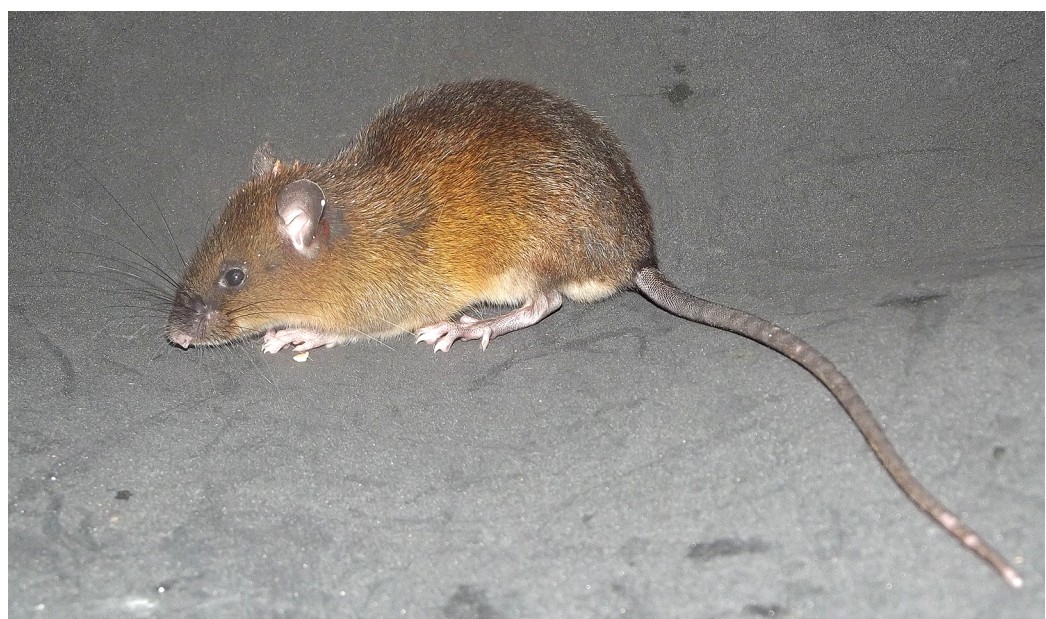

**Figure 9 *Pattonimus ecominga* sp. nov.** *Pattonimus ecominga* sp. nov. (MECN 5928, holotype), an adult male from Reserva Drácula, Carchi, Ecuador.

(boneless autopodium) and skin in regular condition. All of them were collected in Gualpi, Km. 18 of the Gualpi road (0.853841°N, 78.237600°W, 2,350 m) by H. Yela, J. Robayo, and J. Brito on 12 May 2018. MECN 4991 (JBM 1310), a young female; preserved as skull, with carcass and viscera in ethanol; collected at Km. 14 of the Gualpi road (0.882408°N, 78.223235°W, 1,970 m) by J. Brito, J. Robayo, L. Recalde, T. Recalde, and C. Reyes on 5 June 2016.

**Type locality:** Gualpilal (0.891944°N, 78.20308°W, [coordinates taken by GPS at the trap site], elevation 1,700 m), Km. 12 of the Gualpi road, Reserva Drácula, Parroquia Chical, Canton Tulcán, Provincia Carchi, República del Ecuador.

**Diagnosis:** A species of *Pattonimus* gen. nov. with antorbital bridge dorsally broadened, alar fissure typically without a basal notch, a small contribution of parietals in the lateral view, upper incisors with enamel cream or white-colored, and molar occlusal topography moderately complex including mesolophs in M1–M2.

**Morphological description of the holotype and variation:** Dorsal fur reddish brown with a subtle darker middorsal stripe; flanks tending to more reddish (Fig. 9); ventral pelage grayish (Supplemental Information S11); tail long and unicolored (dark above and below), some specimens are slightly paler below. Cranium with moderately long and wide rostrum (Fig. 10); rostral sides taper gradually forward from nasolacrimal capsules; nasals gradually divergent forward with distal ends moderately upturned; shallow but distinct zygomatic notches; interorbit wide, anteriorly convergent with sharp supraorbital margins; fronto-parietal suture V-shaped; braincase moderately inflated and elongated; cranial roof dorsal profile flat from nasals to the half of parietals to slope sharply

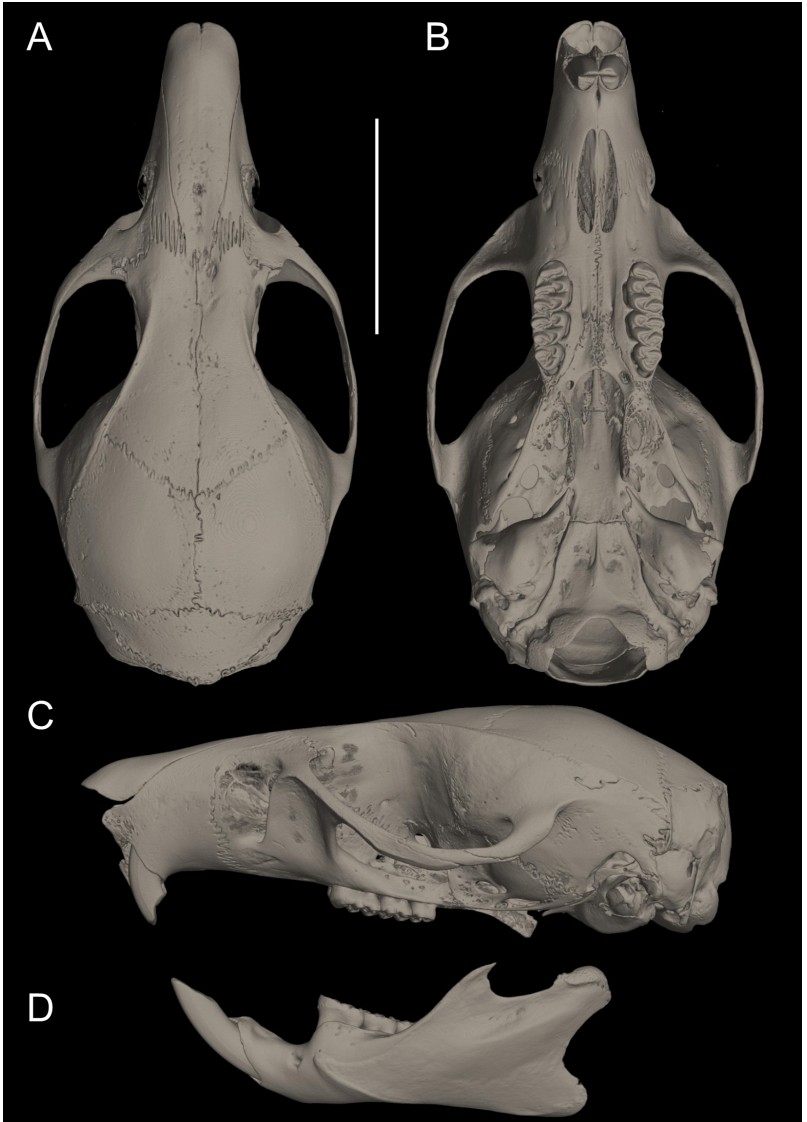

**Figure 10 Cranium in dorsal, ventral, and lateral views, and mandible in labial view.** *Pattonimus ecominga* sp. nov. (Reserva Drácula, Carchi, Ecuador): (A) cranium in dorsal, (B) ventral, (C) lateral views, and (D) mandible in labial view (MECN 5928, holotype). Scale = 10 mm.

downward toward the occiput; foramen magnum oriented posteroventrad; premaxillae slightly shorter than nasals, not produced anteriorly beyond incisors to form a rostral tube; gnathic process small but distinct; zygomatic plate broad and excavated, its anterior edge slightly sloping backwards; zygomatic arches sturdy and robust; maxillary extension of the zygomatic arch with a projection in its lower border; squamosal-alisphenoid groove poorly visible through the translucent braincase, without a perforation where it crosses the depression for the masticatory nerve; small stapedial foramen and carotid canal and barely expressed petrotympanic fissure; primitive cephalic arterial supply (pattern 1 of *Voss, 1988*); alisphenoid strut consistently present, separating buccinator-masticatory

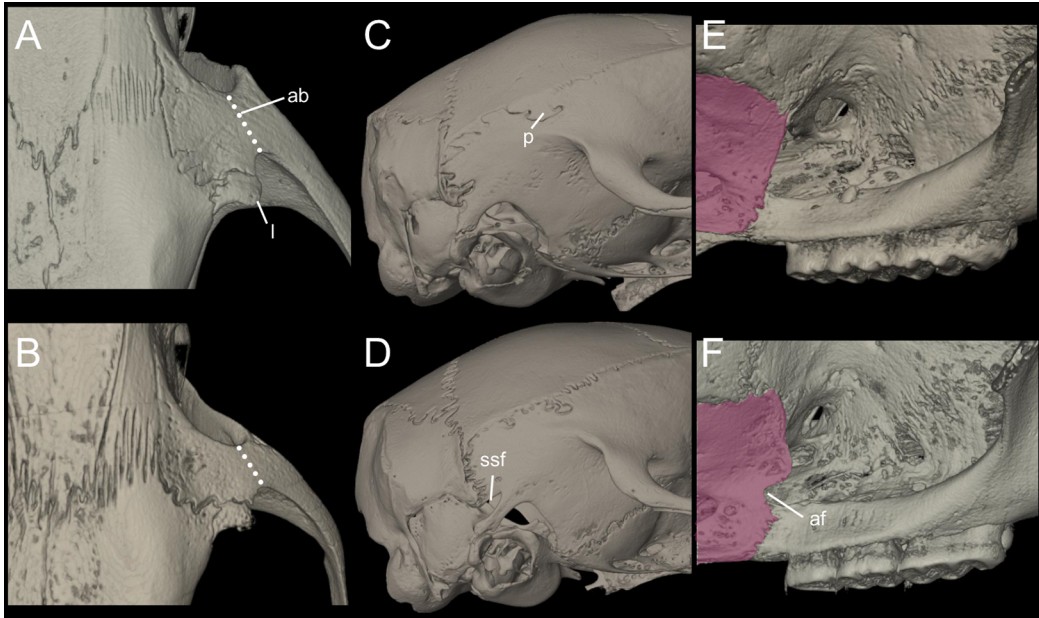

**Figure 11 Main cranial traits differentiating species of *Pattonimus* gen. nov.: *Pattonimus ecominga* sp. nov. (top; MECN 5928, holotype) vs *Pattonimus musseri* sp. nov.** Main cranial traits differentiating species of *Pattonimus* gen. nov.: *Pattonimus ecominga* sp. nov. (A, C and E; MECN 5928, holotype) vs *Pattonimus musseri* sp. nov. (B, D and F; MEPN 12605, holotype). A, B, zygomatic notch region in dorsal view; (C and D) right posterior part of the cranium in lateral view; and (E and F) right orbital region in lateral view (zygomatic arch removed). Abbreviations: ab, antorbital bridge; af, alar fissure (with a basal notch); l, lacrimal; p, parietal (lateral expression); ssf, subsquamosal fenestra.

foramen and foramen ovale; large anterior opening of alisphenoid canal; postglenoid foramen narrow separated from an also narrow subsquamosal fenestra by short and broad hamular process of squamosal (Fig. 11); incisive foramina short, teardrop-shaped, well anterior to the M1s anterior faces; capsular process of premaxillary well developed; palate narrow and short; with the anterior border of the mesopterygoid fossa defined by M3s posterior faces; small posterolateral pits paired and located side by side to the anterior part of the mesopterygoid fossa; squared and short hamular processes; mandible robust; inferior masseteric ridge well-marked; upper incisors with cream enamel bands and straight dentine fissures. Maxillary molar rows slightly divergent backwards; upper molars large, with tendency to lamination and moderate hypsodonty; coronal surfaces slightly crested; main cusps slightly alternated and sloping backwards when viewed from side; M1 subrectangular in outline with procingulum not divided into labial and lingual conules, anteriorly-posteriorly compressed, without anteromedian flexus; mesoloph present; posteroloph present as a small fossete; M2 squared in outline but posteriorly compressed with a procingulum limited to a labial loph; mesoloph, mesostyle, and posteroloph showing the same condition as in M1; M3 subtriangular in outline with an inconspicuous hypoflexus and a compressed posterior lobe (Fig. 12); mandibular molars with main cusps alternated and sloping backwards when viewed from side; procingulum of m1 not divided into labial and lingual conulids; metaflexid fused with the protoflexid; metaconid

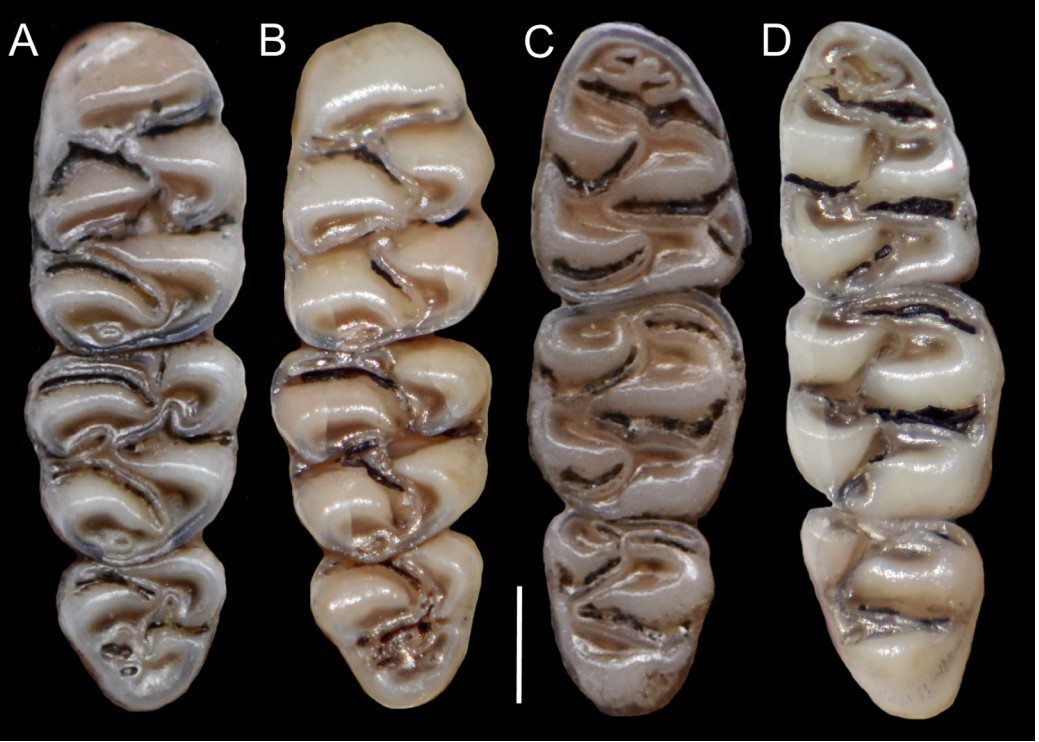

**Figure 12 Upper and lower right toothrows.** (A and B) Upper and (C and D) lower right toothrows in occlusal view of *Pattonimus ecominga* sp. nov. (A and C; Reserva Drácula, Carchi, Ecuador; MECN 5928, holotype) and *Pattonimus musseri* sp. nov. (B and D; Reserva Río Manduriacu, Imbabura, Ecuador; MEPN 12605, holotype). Scale = 1 mm.

connected to protoconid through a narrow bridge; anterolophid indistinct; mesolophid absent; posterolophid present and large; m2 squared in outline; mesolophid and posterolophid showing the same condition as in m1; m3 triangular in outline with a deep hypoflexid and a compressed posterior lobe; enlarged cingulum anterolabial (Fig. 12); gross stomach configuration unilocular-hemiglandular (Fig. 8); gall bladder absent; three diastemal and seven interdentals palatal rugae; the interdental palatal rugae 2–7 with jagged anterior edges.

**Measurements (in mm, except body mass) of the holotype:** Head and body length = 145, tail length = 180, hind foot length = 37, ear length = 17, length of longest mystacial vibrissae = 52, length of longest superciliary vibrissae = 32, length of longest genal vibrissae = 20, body mass = 18 g, occipitonasal length = 34.31, condylo-incisive length = 31.87, length of upper diastema = 8.9, crown length of maxillary toothrow = 5.63, length of incisive foramen = 4.98, breadth of incisive foramina = 1.84, breadth of M1 = 1.73, breadth of rostrum = 6.3, length of nasals = 12.31, length of palatal bridge = 7.31, breadth of bony palate = 2.76, least interorbital breadth = 5.68, zygomatic breadth = 17.2, breadth of zygomatic plate = 3.87, lambdoidal breadth = 12.87, orbital fossa length = 11.12, bular breadth = 3.98, length of mandible = 18.62, crown length of mandibular toothrow = 5.59, length of lower diastema = 4.06, length M1 = 2.72, width M1 = 1.77,

**Table 4 Individual molar measurements (in mm).**

| Collection | P. ecominga sp. nov. | | | | | | | | | | | P. musseri sp. nov. | | | |
|---|---|---|---|---|---|---|---|---|---|---|---|---|---|---|---|
| | MECN | MECN | MECN | MECN | MECN | MECN | MECN | MECN | MECN | MECN | MENC | MEPN | MEPN | MEPN | MEPN |
| Number | 5927 | 5928* | 6017 | 6019 | 6020 | 6025 | 6040 | 6041 | 6042 | 6043 | 6173 | 12586 | 12587 | 12593 | 12605* |
| Age | 3 | 3 | 1 | 3 | 3 | 4 | 3 | 3 | 3 | 3 | 3 | 3 | 5 | 0 | 4 |
| Length M1 | 2.54 | 2.72 | 2.61 | 2.80 | 2.79 | 2.64 | 2.86 | 2.78 | 2.79 | 2.50 | 2.87 | 2.47 | 2.74 | 2.68 | 2.65 |
| Width M1 | 1.73 | 1.77 | 1.80 | 1.77 | 1.73 | 1.74 | 1.79 | 1.75 | 1.74 | 1.67 | 1.79 | 1.68 | 1.80 | 1.73 | 1.70 |
| Length M2 | 1.51 | 1.75 | 1.74 | 1.56 | 1.46 | 1.45 | 1.62 | 1.52 | 1.60 | 1.75 | 1.59 | 1.31 | 1.58 | 1.76 | 1.48 |
| Width M2 | 1.75 | 1.78 | 1.64 | 1.84 | 1.68 | 1.68 | 1.78 | 1.73 | 1.65 | 1.70 | 1.73 | 1.77 | 1.88 | 1.80 | 1.75 |
| Length M3 | 1.30 | 1.24 | – | 1.17 | 1.23 | 1.15 | 1.10 | 1.35 | 1.35 | 1.13 | 1.25 | 1.23 | 1.39 | – | 1.25 |
| Width M3 | 1.37 | 1.35 | – | 1.36 | 1.33 | 1.32 | 1.30 | 1.30 | 1.32 | 1.33 | 1.32 | 1.34 | 1.42 | – | 1.41 |
| Length m1 | 2.25 | 2.28 | 2.16 | 2.38 | 2.41 | 2.35 | 2.10 | 2.30 | 2.26 | 2.21 | 2.33 | 2.16 | 2.28 | 2.23 | 1.94 |
| Width m1 | 1.66 | 1.66 | 1.60 | 1.68 | 1.69 | 1.74 | 1.57 | 1.74 | 1.60 | 1.57 | 1.66 | 1.66 | 1.70 | 1.70 | 1.50 |
| Length m2 | 1.71 | 1.72 | 1.77 | 1.76 | 1.76 | 1.66 | 1.75 | 1.68 | 1.70 | 1.73 | 1.74 | 1.56 | 1.80 | 1.76 | 1.76 |
| Width m2 | 1.76 | 1.68 | 1.67 | 1.76 | 1.67 | 1.71 | 1.70 | 1.73 | 1.66 | 1.55 | 1.73 | 1.68 | 1.81 | 1.78 | 1.67 |
| Length m3 | 1.55 | 1.47 | – | 1.54 | 1.65 | 1.61 | 1.56 | 1.65 | 1.62 | 1.41 | 1.52 | 1.62 | 1.70 | – | 1.43 |
| Width m3 | 1.45 | 1.34 | – | 1.46 | 1.35 | 1.34 | 1.34 | 1.36 | 1.36 | 1.30 | 1.37 | 1.35 | 1.44 | – | 1.38 |

Notes:
 * Holotypes.
 Individual molar measurements (in mm) of the type series of *Pattonimus ecominga* sp. nov. and *Pattonimus musseri* sp. nov. (Oryzomyini, Sigmodontinae).

length M2 = 1.75, width M2 = 1.78, length M3 = 1.24, width M3 = 1.35, length m1 = 2.28, width m1 = 1.66, length m2 = 1.72, width m2 = 1.68, length m3 = 1.47, width m3 = 1.34. Measurements for the paratypes are given in Tables 3 and 4.

**Distribution:** Known from several neighboring collecting sites in Reserva Drácula (Carchi, Ecuador), on the western flank of the Andes (Fig. 6), at an elevation of 1,600–2,340 m.

**Natural history:** Reserva Drácula is located in the headwaters of the río Gualpi in the subtropical and lower montane ecosystem (*Cerón et al., 1999*). The local expression of the cloud montane forest is characterized by a tree canopy that reaches 30 m high; the understory is luxurious and mostly composed of species belonging to Araceae, Melastomataceae, Cyclanthaceae, Bromeliaceae, and ferns (Supplemental Information S12). A recently captured specimen showed a calm behavior, foraging on the ground between the roots (Supplemental Information S13), where we observed it feeding on small seeds. From the same pit falls where *P. ecominga* sp. nov. was obtained, we also collected the sigmodontines *Chilomys* sp., *Melanomys caliginosus*, *Microryzomys minutus*, *Nephelomys* cf. *pectoralis*, *Oecomys* sp., *Rhipidomys latimanus*, *Tanyuromys thomasleei*, and *Thomasomys bombycinus*, the heteromyid *Heteromys australis*, the marsupials *Caenolestes convelatus*, *Mamosops caucae*, and *Marmosa* sp., and the soricid *Cryptotis equatoris*.

**Etymology:** The specific name is the Spanish name "ecominga;" it honours the NGO Fundación EcoMinga, an Ecuadorian foundation with international sponsors, focused on the conservation of the unique foothill forests, cloud forests, and alpine grasslands ("páramo") of the Andes, especially those on the edge of the Amazon basin in east-central

Ecuador and those on the super-wet western Andean slopes of the Chocó region in northwest Ecuador.

**Conservation:** Most parts of the Reserva Drácula are primary forests that have never been cut (at least according to the historical records). However, significant portions of this forest have recently been cleared along the road to establish fields of "naranjilla" plantations (*Solanum quitoense*), a fruit of high commercial value. This plant produces good crops for 2 years. After that, the soil becomes contaminated with pathogens and pesticides, so the cultivation is no longer profitable, and these old fields are abandoned or used as pasture.

*Pattonimus musseri* sp. nov.
urn:lsid:zoobank.org:act:A50ABD02-60BA-497C-9DCE-83D6C7811305
Musser's montane rat, Rata montana de Musser

**Holotype**: MEPN 12605 (field number JBM 1752), an adult female represented by a skull and partial postcranial skeleton and skin in good condition; collected by J. Brito and Glenda Pozo on 12 April 2017.

**Paratopotypes:** MEPN 12586 (JBM 1733), an adult female; MEPN 12593 (JBM 1740), a young male; and MEPN 12587 (JBM 1734), an adult male; all preserved as skulls, partial postcranial skeletons and museum skins in regular conditions and collected by J. Brito and G. Pozo between 12 and 14 April 2017.

**Type locality:** Reserva Río Manduriacu (0.309547°N, 78.856631°W, [coordinates taken by GPS at the trap site], elevation 1,200 m), Parroquia García Moreno, Cantón Cotacachi, Provincia Imbabura, República del Ecuador.

**Diagnosis:** A species of *Pattonimus* gen. nov. with antorbital bridge dorsally narrow, alar fissure with a basal notch, lateral expression of parietal absent, upper incisors with enamel orange-colored, and molar occlusal topography simplified, typically lacking mesolophs on M1–M2.

**Morphological description of the holotype and variation:** Dorsal fur reddish brown with a darker middorsal stripe; flanks tending to more reddish; ventral pelage grayish (Supplemental Information 14); tail long and unicolored (dark above and below). Cranium with moderately long and wide rostrum; rostral sides taper gradually forward from nasolacrimal capsules (Fig. 13); nasals gradually divergent forward with distal ends moderately upturned; shallow but distinct zygomatic notches; with antorbital bridge dorsally narrowed (Fig. 11); interorbit wide, anteriorly convergent with sharp supraorbital margins; fronto-parietal suture U-shaped; braincase moderately inflated and elongated; cranial roof dorsal profile flat from nasals to the half of parietals to slope sharply downward toward the occiput; foramen magnum oriented posteroventrad; premaxillae slightly shorter than nasals, not produced anteriorly beyond incisors to form a rostral tube; gnathic process small but distinct; zygomatic plate broad and excavated, its anterior edge slightly sloping backwards; zygomatic arches sturdy and robust; maxillary extension of the zygomatic arch with projection in its forward border; squamosal-alisphenoid groove

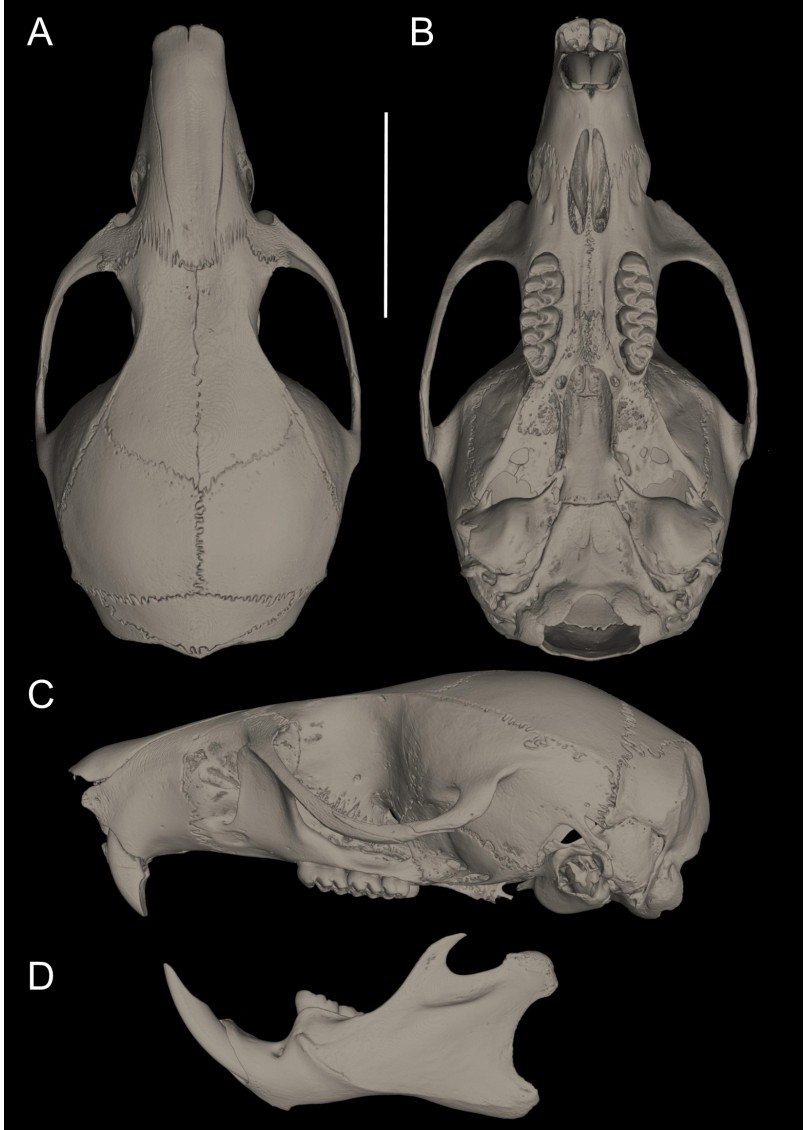

**Figure 13 *Pattonimus musseri* sp. nov.** *Pattonimus musseri* sp. nov. (Reserva Río Manduriacu, Imbabura, Ecuador): (A) cranium in dorsal, (B) ventral, (C) lateral view, and (D) mandible in labial views (MEPN 12605, holotype). Scale = 10 mm.               

poorly visible through the translucent braincase, without a perforation where it crosses the depression for the masticatory nerve; small stapedial foramen and carotid canal but barely expressed petrotympanic fissure; primitive cephalic arterial supply (pattern 1 of *Voss, 1988*); alisphenoid strut consistently present, separating buccinator-masticatory foramen and foramen ovale; small anterior opening of alisphenoid canal; alar fissure with a basal notch, lateral expression of parietal absent (Fig. 11); postglenoid foramen narrow, subsquamosal fenestra small, narrow and long hamular process of squamosal; square tegmen tympani. Incisive foramina short, teardrop-shaped, well anterior to the M1s anterior faces; capsular process of premaxillary well developed; palate narrow and short;

with the anterior border of the mesopterygoid fossa defined by M3s posterior faces; small posterolateral pits paired and located side by side to the anterior part of the mesopterygoid fossa; squared and short hamular processes; petrosal little exposed (Fig. 13); mandible robust, with the vertical branch straight (Fig. 13); inferior masseteric ridge well-marked; upper incisors with enamel orange-colored and straight dentine fissure; maxillary molar rows slightly divergent backwards; upper molars large, with tendency to lamination and moderate hypsodonty; coronal surfaces slightly crested; main cusps slightly alternated and sloping backwards when viewed from side; M1 subrectangular in outline with procingulum not divided into labial and lingual conules, anteriorly-posteriorly compressed, without anteromedian flexus; mesoloph absent (Fig. 12); posteroloph present as a small fossete; M2 squared in outline but posteriorly compressed with a procingulum limited to a labial loph; mesoloph, mesostyle, and posteroloph showing the same condition as in M1; M3 subtriangular in outline with an inconspicuous hypoflexus and a compressed posterior lobe; mandibular molars with main cusps alternate and sloping backwards when viewed from side. Procingulum of m1 not divided into labial and lingual conulids; metaflexid fused with the protoflexid; metaconid connected to protoconid through a narrow bridge; anterolophid indistinct; mesolophid absent (Fig. 12); posterolophid present and large; m2 squared in outline; mesolophid and posterolophid showing the same condition as in m1; m3 triangular in outline with a deep hypoflexid and a compressed posterior lobe.

**Measurements (in mm, except body mass) of the holotype:** Head and body length = 140, tail length = 177, hind foot length = 35, ear length = 19, length of longest mystacial vibrissae = 49.17, length of longest superciliary vibrissae = 28.42, length of longest genal vibrissae = 19.14, body mass = 59 g, occipitonasal length = 31.2, condylo-incisive length = 29.05, length of upper diastema = 7.96, crown length of maxillary toothrow = 5.56, length of incisive foramen = 4.72, breadth of incisive foramina = 1.73, breadth of M1 = 1.7, breadth of rostrum = 6.03, length of nasals = 12.03, length of palatal bridge = 6.77, breadth of bony palate = 2.43, least interorbital breadth = 5.68, zygomatic breadth = 16.51, breadth of zygomatic plate = 3.49, lambdoidal breadth = 13.1, orbital fossa length = 10.06, bular breadth = 3.75, length of mandible = 16.93, crown length of mandibular toothrow = 5.31, length of lower diastema = 4.17, length M1 = 2.65, width M1 = 1.70, length M2 = 1.48, width M2 = 1.75, length M3 = 1.25, width M3 = 1.41, length m1 = 1.94, width m1 = 1.50, length m2 = 1.76, width m2 = 1.67, length m3 = 1.43, width m3 = 1.38. Measurements for the paratypes are given in Tables 3 and 4.

**Etymology:** This species is named in honour of Guy G. Musser (1936–2019), outstanding collector and taxonomist devoted to the study of worldwide muroid rodents (*Carleton, 2009*). We adopted as ours what *Voss & Carleton (2009*: 3*)* wrote about Musser's legacy, "his publications set new standards in systematic mammalogy." The species epithet is formed from the surname "Musser," taken as a noun in the genitive case, with the Latin suffix "i" (ICZN 31.1.2).

**Distribution:** Restricted to the type locality on the western flank of the Andes, Reserve Río Manduriacu, Imbabura province, Ecuador (Fig. 6), at an elevation of 1,200 m.

**Natural history:** Reserva Río Manduriacu is placed in the headwaters of the Manduriacu River, a region belonging to the subtropical ecosystem (*Albuja et al., 2012*). The vegetation corresponds to the Low Montane Evergreen Forest of the western slopes of the Andes (*Cerón et al., 1999*). From the same pit falls where *P. musseri* sp. nov. was obtained, we also collected the sigmodontines *Neacomys tenuipes*, *Melanomys caliginosus*, *Tanyuromys thomasleei*, *Transandinomys bolivaris*, the heteromyid *Heteromys australis*, and the marsupial *Mamosops caucae*.

**Conservation:** The reserves Río Manduriacu and Drácula are threatened by the expansion of mining concessions across the northwest of Ecuador (*Roy et al., 2018*; *Guayasamin et al., 2019*). The western Andean slopes from Ecuador (Chocó Region) have shown important micro-regions of small vertebrate endemism, which are restricted to areas with good-quality forest and very little or no anthropogenic activity (*Yánez-Muñoz et al., 2018*; *Guayasamin et al., 2019*). Thus, activities that threaten these Chocó forests must be regulated and authorized within the framework of the Ecuadorian Constitution. A program of conservation actions for biodiversity is also needed for the Ecuadorian Andes. Such program have advanced mostly with the participation of non-profit institutions that aim to protect priority and vulnerable forests for biodiversity conservation, such as those carried out by the Fundación EcoMinga (*Yánez-Muñoz et al., 2018*; *Guayasamin et al., 2019*).

*Pattonimus* sp.
**Referred material:** QCAZ 8720, preserved as skull (Supplemental Information S15) and body in fluid, collected at Otonga (0.4189°N, 79.0039°W, 2,065 m), Provincia Cotopaxi, Ecuador (*Pinto et al., 2018*); ICN 13663 and ICN 21487, preserved as skulls and skins, collected at the Fundación Ecológica Los Colibries de Altaquer (1.293111°N, 78.073972°W, 1,100 m), Reserva del río Ñambi, Corregimiento Altaquer, Municipio de Barbacoas, Departamento Nariño, Colombia (*Cadena, Anderson & Rivas-Pava, 1998*).

**Remarks:** More field work is necessary in Reserva Otonga (Ecuador) and in the Reserva del Río Ñambi (Colombia), in order to collect additional material that allows exploring both morphology and genetics to properly allocate these populations.

## DISCUSSION

**Pattonimus gen. nov. molar morphology in oryzomyine dental morphospace:**
The recognition of *Pattonimus* gen. nov. as a distinct genus, that is, a separate evolutionary lineage occupying a unique biogeographic and ecological zone, is supported by several pieces of information, including molecular data and integumental and cranial characters. Nevertheless, it is the unique molar morphology of *Pattonimus* gen. nov. among oryzomyines that provides the best evidence for the ecological significance of this taxon. To our perception, this genus represents a novel transition to a dental morphospace within

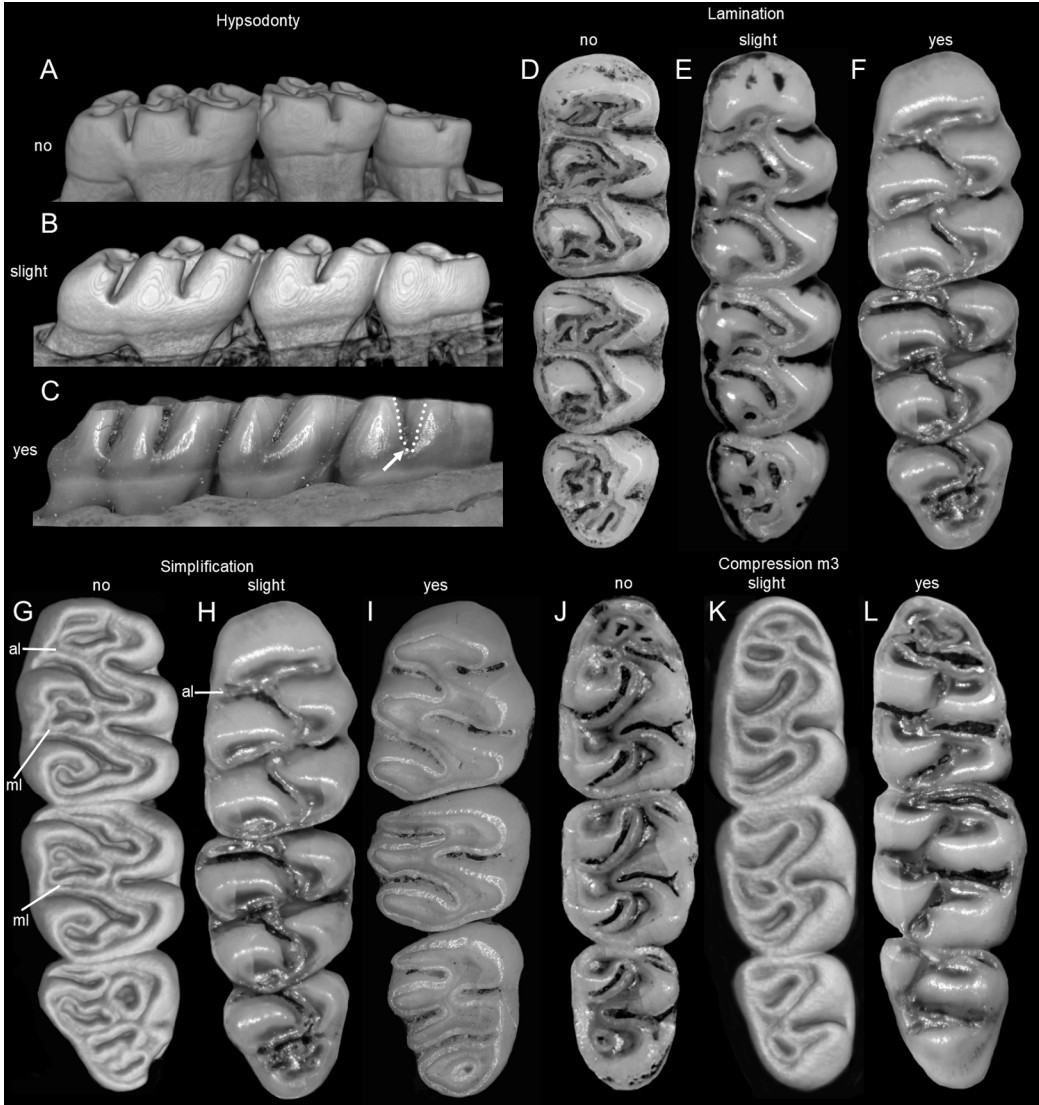

**Figure 14 Four selected traits discussed in the main text illustrating molar variability in extinct (†) and extant oryzomyines.** Four selected traits discussed in the main text illustrating molar variability in extinct (†) and extant oryzomyines. Hypsodonty ((A) ZFMK 2016-0981-sk, †*Megaoryzomys curioi*; (B) MEPN 12605, *Pattonimus musseri* sp. nov.; (C) CNP-E 882-2, *Holochilus chacarius*); lamination ((D) BMNH 13.10.24.58, *Mindomys hammondi*; (E) MEPN 11719, *Transandinomys bolivaris*; (F) MEPN 12605, *P. musseri* sp. nov.); simplification ((G) MECN 3407, *Tanyuromys thomasleei*; (H) MEPN 12605, *P. musseri* sp. nov.; (I) CNP 3964 *Holochilus chacarius*); m3 compression ((J) MECN 3797, *Nephelomys auriventer*; (K) MECN 6021, *Sigmodontomys alfari*; (L) MEPN 12605, *P. musseri* sp. nov.). Abbreviations: al, anteroloph, ml, mesoloph. The arrow indicates the depth of the hypoflexus of the third molar, which denotes the hypsodonty.                   

the tribe that combines lamination, increased crown height (i.e., relatively more hypsodont), occlusal simplification, and a mesiodistally compressed m3 (Fig. 14).

Molar morphology, including tooth proportions, crown height, and occlusal topography, shows important variation within Oryzomyini (*Musser et al., 1998*; *Weksler, 2006*). This is not unexpected, since this tribe has a noticeable taxonomic diversity (40 genera including both extinct and extant genera), and displays significant variation
in body sizes, diets, life modes, and biomes colonized (*Voss, 1991*; *Carleton & Musser, 1989*; *Musser et al., 1998*; *Weksler, 2006*). Nevertheless, few forms depart from a "typical" oryzomyine molar bauplan, recognized as early as *Hershkovitz (1944)*: brachydont, bunodont, and pentalophodont. *Pattonimus* gen. nov. is unique in this regard, and its dentition deserves further consideration.

   *Pattonimus* gen. nov. molars have relatively higher crowns than that of most oryzomyines. Hypsodonty "… *is the evolutionary process that provides a longer wearing surface by an increase in the depth of the tooth*" (*Hershkovitz, 1962*: 88). If the classical definition of hypsodonty is used, that is, cheek tooth crown height exceeding its anteroposterior length (*Williams & Kay, 2001*), no oryzomyine can be considered as hypsodont and, in fact, oryzomyines are typically treated as brachydont sigmodontines (e.g., *Hershkovitz, 1960*; *Prado & Percequillo, 2018*; *Turvey et al., 2010*; *Musser et al., 1998*). *Weksler (2006*: 44*)* indicated that just "…*the molars of* Holochilus *are hypsodont… Remaining [extant] oryzomyines have bunodont and brachyodont molars.*" Nevertheless, it is clear that there is a considerable degree of variation in crown elongation among members of the tribe, and several studies have used the term hypsodont in a comparative sense. For example, *Carleton & Olson (1999*: 25*)* discussed the hypsodonty of the extinct *Noronhomys*, against that of *Holochilus*, indicating "*The dissimilarity in closure of the lingual folds may relate to the greater coronal hypsodonty seemingly characteristic of* Noronhomys.*" *Pardiñas (2008*: Table 2*)*, listed the genera *Carletonomys*, *Noronhomys*, and *Holochilus* as hypsodont, but considered *Pseudoryzomys* and *Lundomys* as "higher crowned," and, by this action, highlighted the existence of some degree of variation in hypsodonty within the tribe. When describing *Drymoreomys*, *Percequillo, Weksler & Costa (2011*: 365*)* stated that the genus has "… *labial and lingual cusps high (molar nearly hypsodont).*" *Pine, Timm & Weksler (2012*: 862*)* indicated that "Tanyuromys *differs from both* Nectomys *and* Sigmodontomys *in having much more complex molar patterns, less-hypsodont molars.*" In fact, the tendency of *Nectomys* to have high-crowned molars, in comparison with other oryzomyines, is largely recognized (see *Ellerman, 1941*: 361; *Hershkovitz, 1944*: 19). In *Mindomys*, according to *Percequillo (2015*: 360*)*, "*the molars are pentalophodont and moderately high-crowned.*" Comparative studies employing quantitative measures are necessary to infer crown elongation among Oryzomyini, which apparently represents a gradient of conditions and resists simplistic approaches, although, as was acutely highlighted by *Carleton & Olson (1999*: 25*)*, "…*an impression* [the hypsodonty variation] *that we cannot easily quantify, however.*"

   According to *Hershkovitz (1962*: 92*)*, lamination "…*is the process of transection of a molar crown by confluence of a fold of one side of the tooth with another of the opposite side.*" Technically, full lamination was not achieved within Cricetidae (cf. *Ellerman, 1941*; *Stehlin & Schaub, 1951*; *Hershkovitz, 1962*; *Vorontsov, 1967*), although an important degree of transverse lamination (i.e., confluence of directly opposing folds; *Hershkovitz, 1962*: 92*)* is observed in a few taxa, such as the sigmodontine *Irenomys* (*Teta & Pardiñas, 2015*). A tendency to lamination is also recognized in several oryzomyines. *Voss, Gómez-Laverde & Pacheco (2002*: 15*)*, describing *Handleyomys*, stated that "*the principal labial flexi …slant transversely across the midline of the tooth to interpenetrate*

*with much longer lingual flexi, resulting in the morphology that* Voss (1993*: 20) termed 'incipient lophodonty'* [in reference to the sigmodontine genus *Delomys*]." The lophodont condition of a molar refers to the presence of ridges or lophs interconnecting the cusps; in many cases, lophs acquire the form of laminae (Mones, 1979). For instance, the Otomyinae murids are characterized by an extreme lophodonty, distinguished with the specific term loxodonty, displaying molars composed of numerous laminae or prisms achieved via lamination (Denys, Michaux & Hendey, 1987). Within Oryzomyini, lamination reaches its high known expression in the fossil *Noronhomys* and in some species of *Holochilus* (e.g., *H. sciureus*; Massoia, 1976; Voss & Carleton, 1993; Carleton & Olson, 1999). The condition of the lamination displayed by these oryzomyines is what Hershkovitz (1962: 93) described as "*oblique* [lamination]… *confluence of a fold of one side with either the anterior or posterior alternating fold of the opposite side.*"

Hypsodonty in sigmodontines is usually linked with planar occlusal surfaces and simplification (see Ronez et al., 2020a, and the references cited therein). The latter process constituted one of the main elements in Hershkovitz (1962) understanding about molar sigmodontine evolution which involved, almost axiomatically, the evolutionary transition from complex (pentalophodont) to secondarily simplified (tetralophodont or derivates) molars. Simplification implies the loss or obsolescence of occlusal structures, particularly the complex mesoloph-mesostyle (i.e., tetralophodont molars), and also additional crests (e.g., anteroloph, posteroloph; Hershkovitz, 1962: 76). Historically, Oryzomyini were treated as mostly pentalophodont sigmodontines (Weksler, 2006, and references cited therein), but the phylogenetic allocation of several tetralophodont genera as oryzomyines, namely *Holochilus*, *Lundomys*, *Pseudoryzomys*, and *Zygodontomys* (Voss & Carleton, 1993; Weksler, 2006) compromised this traditional concept. The set of dentally simplified oryzomyines also includes the fossil taxa *Carletonomys*, *Noronhomys*, and *Reigomys* (Machado et al., 2014). All phylogenetic evidence to date, including our results, points out that molar simplification operated in at least two main lineages within the tribe: (1) in *Zygodontomys*, which is not closely related to the remaining taxa and is placed at the base of oryzomyine diversification; and (2) in a clade containing *Holochilus*, *Lundomys*, *Pseudoryzomys*, and the above mentioned fossil taxa, which have been recovered consistently grouped (Carleton & Olson, 1999; Weksler, 2006; Machado et al., 2014). We propose here that *Pattonimus* gen. nov. represents an additional oryzomyine lineage that is undergoing a morphological transition to a simplified occlusal surface, coupled with incipient lamination, hypsodonty, and m3 compression.

In summary, we are convinced that the unique combination of dental traits displayed by *Pattonimus* gen. nov. deserves generic recognition and that molar morphology diversity within oryzoymines is markedly enlarged. Other arguable classificatory schemes could be to consider these forms as members of already established genera such as *Mindomys* or *Nephelomys*. However, this latter alternative hypothesis implies the acceptance that these taxa embrace an extreme range of variability in the occlusal design of their molars. Speciose genera within Oryzomyini, such as *Cerradomys*, *Neacomys*, the *Nephelomys* sensu stricto, or *Oecomys*, are markedly conservative in molar morphology (Tavares, Pessôa & Gonçalves, 2011; Bonvicino, Casado & Weksler, 2014; Hurtado & Pacheco, 2017; Musser

*et al., 1998*; *Pardiñas et al., 2016*), which constitutes an accessory support to our preferred hypothesis.

**Oryzomyine diversification in northern Andes and the aggregated value of *Pattonimus* gen. nov.:** The northern Andes have been long highlighted as an important region for the diversification of the tribe Oryzomyini. The most significant contribution on this topic, prior to the popularization of phylogenetic analysis, was based on the patterns of species richness in South America, conducted by *Reig (1986)*. Evaluating the species composition of the tribes of Sigmodontinae, he pointed out that the northern Andes were the "area of original differentiation" for the oryzomyines, a region from where this group originated and dispersed throughout the continent. As outlined by *Prado & Percequillo (2013)*, the composition of the tribe at that time was quite diverse, including several genera now assigned to the tribe Thomasomyini, and much of the diversity that has since been recognized.

In fact, the northern portion of the Andean cordillera houses an incredible diversity of oryzomyine genera, such as *Aegialomys*, *Handleyomys*, "*Handleyomys*" (species of the alfaroi group; see *Weksler, 2015*), *Melanomys*, *Mindomys*, *Microryzomys*, *Nephelomys*, *Oreoryzomys*, and now *Pattonimus* gen. nov. Most of these lineages are considered as independent colonizers of the Andes, as they belong to different clades within the tribe and several of them do not share common histories, suggesting that dispersion is the most important process of tribal diversification in this region (*Schenk & Steppan, 2018*). Nevertheless, the phylogenetic relationships recovered here, with *Mindomys*, *Nephelomys* and *Pattonimus* gen. nov. sharing a common ancestor within clade B, suggest that their generic and specific diversification took place locally. This clade would be a truly and unique Andean autochthonous radiation within Oryzomyini, with several species (*Nephelomys*, 12 species; *Mindomys*, one species; *Pattonimus* gen. nov., 2 to 4 species) evolving within these montane forests. Also, considering clade C, it is likely that the ancestor of *Oreoryzomys* and *Microryzomys* colonized this region once, but these genera are poorly diversified (three species only comprising both genera). This interesting issue deserves further exploration, but prima facie is not limited to oryzomyines. In fact, Ichthyomyini, one of the most singular expressions of the sigmodontine radiation, appear as a primary autochthonous Andean radiation in northern South America (*Voss, 1988*) and it is likely that the same happened within the tribe Thomasomyini (*Pacheco, 2015*).

***Pattonimus* gen. nov. and overlooked sigmodontine diversity in northern Andes:** In a worldwide appraisal to current mammalogy research, *Ceballos & Ehrlich (2009*: 3*)* highlighted that "*It appears that exploration of new regions has been the main factor for the discovery of as much as 40% of the new species, such as the pygmy deer (*Muntiacus putaoensis*) in Bhutan, the Arunachal macaque (*Macaca muzala*) from the Himalaya foothills of northeast India, the Amazonian basin monkeys, and most of the new Philippines species… The exploration of new regions has been based on both the use of either new techniques… or traditional techniques, such as pitfall traps, which have yielded specimens of 8 new species of shrew-tenrecs from Madagascar since 1988.*" The case of *Pattonimus* gen. nov. is a suitable example of what "traditional techniques" of collection can achieve when applied in unexplored Andean regions. We are dealing with a new genus and maybe four

new species, a noticeable increment for a mammal group understood as moderately well-known (*Patton, Pardiñas & D'Elía, 2015*). Continuous sampling is crucial, even in well sampled areas, as testifies the description of two new genera and species in the Atlantic Forest of São Paulo, Paraná, Santa Catarina and the rocky outcrops of the Cerrado in Minas Gerais, in southeastern Brazil, in the last ten years, namely *Drymeoreomys albimaculatus* (*Percequillo, Weksler & Costa, 2011*) and *Calassomys apicalis* (*Pardiñas et al., 2014*).

After more than two centuries of active mammalogy research (*Tirira, 2014*), intensive field work was conducted in few Ecuadorian places. Examples for those places in the eastern Andes are Papallacta (*Voss, 2003*), Guandera Biological Reserve (*Lee et al., 2015*), Sangay National Park (*Brito & Ojala-Barbour, 2016*), Yacuri National Park (*Lee et al., 2018*); and in the western Andes are Cajas National Park (*Barnett, 1999*), Otonga Reserve (*Jarrín, 2001*; *Pinto et al., 2018*), Pululahua Geobotanical Reserve (*Curay, Romero & Brito, 2019*), and Polylepis Forest (*Ojala-Barbour, Brito & Teska, 2019*). The interest in complementing biodiversity studies has led to expeditions to little-known locations, such as the Reserva Drácula and also triggered revisions of museum specimens. These approaches have retrieved several recent additions to the Ecuadorian sigmodontine fauna (e.g., *Rhagomys longilingua*, see *Medina et al., 2017*; *Amphinectomys savamis*, see *Chiquito & Percequillo, 2016*; *Nectomys saturatus*, see *Chiquito & Percequillo, 2019*), and also led to the description of new biological entities (e.g., *Rhipidomys albujai*, see *Brito et al., 2017*; *Tanyuromys thomasleei*, see *Timm, Pine & Hanson, 2018*; *Thomasomys salazari*, see *Brito et al., 2019*; *Ichthyomys pinei*, see *De Córdova et al., 2020*). In this same way, recent surveys in isolated Ecuadorian mountain systems, such as Cordillera del Cóndor and Kutukú, are revealing several novel species for monotypic (e.g., *Mindomys*) or speciose genera (e.g., *Neacomys*, *Rhipidomys* and *Thomasomys*), which are still in the process of publication. Such flourishing richness surely will reorganize part of our understanding of Neotropical cricetids. This context highlights the urgency to establish rational and comprehensive programs of inventory and collection as well as to improve the access of scholars to these resources.

## CONCLUSIONS

A new genus, *Pattonimus* gen. nov., is added to Oryzomyini. With at least three species, two of them described here (*Pattonimus ecominga* sp. nov. and *P. musseri* sp. nov.), this new cricetid appears as an endemic form of the montane forest of southern Chocó biogeographic region in western Ecuador and Colombia. Phylogenetic analyses based on combined morphological and genetic evidence resolve *Pattonimus* sp. nov. as sister to *Mindomys*, another Chocoan endemic. Molar morphology highlights the singularity of the new genus by combining moderately hypsodont teeth with simplified occlusal design and a distinct tendency to lamination. Since *Pattonimus* sp. nov. is a novel taxon that has emerged from recent field studies, this is a clear indication of our still fragmentary knowledge of rodent communities in the Andes.

## ACKNOWLEDGEMENTS

To J. Robayo, L. Jost, and H. Schneider of Basel Botanical Garden and Rainforest Trust; to the graduate biologists J. Curay, R. Vargas, R. Garcia, C. Bravo, S. Pozo (the "Minion" team), and C. Nivelo, for invaluable field assistance; to the rangers of EcoMinga, especially H. Yela, T. Recalde, J. M. Loaiza, and D. Meneses for their deep efforts with field logistics; to D. Inclán and F. Prieto of INABIO, for their sponsorship and permanent support. N. Cazzaniga guided us in nomenclatorial aspects. C. Galliari helped in multivariate analyses. R. A. Martin generously read the manuscript and corrected the grammar and spelling. We are deeply indebted to the above-mentioned persons and institutions.

### Funding

This work was supported by Fundación EcoMinga (Jorge Brito and Ulyses Pardiñas), Germany-Brazil-Ecuador Trilateral Cooperation Program, funded by the international cooperation GIZ (Jorge Brito and Claudia Koch), Fundação de Amparo à Pesquisa do Estado de São Paulo (FAPESP) 98/12273-0, 09/16009-1, 15/20055-2 and Conselho Nacional de Pesquisa e Desenvolvimento Científico e Tecnológico (CNPq) 305164/2011-2, 307519/2015-5, 476249/2008-2 (Alexandre R. Percequillo) and 308866/2017-7 and 440663/2015-6 (to Marcelo Weksler). The laboratory work in Ecuador were funded by grants from Secretaría de Educación Superior, Ciencia Tecnología e Innovación (SENESCYT), Project "Arca de Noe", Santiago R. Ron and Omar Torres-Carvajal Principal Investigators (Nicolás Tinoco). The funders had no role in study design, data collection and analysis, decision to publish, or preparation of the manuscript.

### Grant Disclosures

The following grant information was disclosed by the authors:
Fundación EcoMinga.
International Cooperation GIZ.
Fundação de Amparo à Pesquisa do Estado de São Paulo (FAPESP): 98/12273-0, 09/16009-1 and 15/20055-2.
Conselho Nacional de Pesquisa e Desenvolvimento Científico e Tecnológico (CNPq): 305164/2011-2, 307519/2015-5 and 476249/2008-2 and 308866/2017-7 and 440663/2015-6.
Secretaría de Educación Superior, Ciencia Tecnología e Innovación (SENESCYT).

### Competing Interests

The authors declare that they have no competing interests.

### Author Contributions

- Jorge Brito conceived and designed the experiments, performed the experiments, analyzed the data, prepared figures and/or tables, authored or reviewed drafts of the paper, acquired the funds, and approved the final draft.

- Claudia Koch performed the experiments, analyzed the data, prepared figures and/or tables, authored or reviewed drafts of the paper, and approved the final draft.
- Alexandre R. Percequillo performed the experiments, analyzed the data, authored or reviewed drafts of the paper, and approved the final draft.
- Nicolás Tinoco performed the experiments, analyzed the data, prepared figures and/or tables, authored or reviewed drafts of the paper, and approved the final draft.
- Marcelo Weksler performed the experiments, analyzed the data, prepared figures and/or tables, authored or reviewed drafts of the paper, and approved the final draft.
- C. Miguel Pinto performed the experiments, analyzed the data, prepared figures and/or tables, authored or reviewed drafts of the paper, and approved the final draft.
- Ulyses F. J. Pardiñas conceived and designed the experiments, performed the experiments, analyzed the data, prepared figures and/or tables, authored or reviewed drafts of the paper, and approved the final draft.

## Animal Ethics

The following information was supplied relating to ethical approvals (i.e., approving body and any reference numbers):

Handling and all activities regarding specimens followed care and use ethical procedures recommended by the American Society of Mammalogists (*Sikes et al., 2016*).

For the use and care of animals, we follow the guidelines of the Ministry of the Environment of Ecuador, through scientific research authorization No 006-2015-IC-FLO-FAU-DPAC MAE and No 003-2019-IC-FLO-FAU-DPAC/MAE.

## Field Study Permissions

The following information was supplied relating to field study approvals (i.e., approving body and any reference numbers):

Field experiments were approved by the Ministerio del Ambiente del Ecuador (permissions number No 006-2015-IC-FLO-FAU-DPAC/MAE and No 003-2019-IC-FLO-FAU-DPAC/MAE).

## DNA Deposition

The following information was supplied regarding the deposition of DNA sequences:

The Cytb sequences are available at GenBank: MT700413 to MT700428, and IRBP MT700429 to MT700436.

The Cytb and IRBP sequences are available in the Supplemental File.

## Data Availability

The morphological character matrix 103, GenBank accession numbers, Cytb and IRBP sequences, and the specimens/material described and reviewed are available in the Supplemental Files.

The Cytb sequences are available at GenBank: MT700413 to MT700428, and IRBP MT700429 to MT700436.

J.P. Carrera (MEPN), S. Burneo (QCAZ), J. Decher (ZFMK), and R. Hutterer (ZFMK) allowed access to the mammal collections under their charge.

The reconstructed image stacks of the CT-scans are available on www.morphdbase.de, direct links to the data of the specimens used herein are as follows:

Pattonimus ecominga sp. nov. (MECN 5928, holotype): www.morphdbase.de/?C_Koch_20200908-S-13.1.

Pattonimus musseri sp. nov. (MEPN 12605, holotype): www.morphdbase.de/?C_Koch_20200908-S-14.1.

Mindomys hammondi (BM 13.10.24.58, holotype): www.morphdbase.de/?C_Koch_20200922-S-16.1.

Nephelomys auriventer (MECN 5812): www.morphdbase.de/?C_Koch_20200922-S-18.1.

Tanyuromys thomasleei (MECN 3407): www.morphdbase.de/?C_Koch_20200908-S-12.1.

†Megaoryzomys curioi (ZFMK 2016-0981-sk), mandible: www.morphdbase.de/?C_Koch_20200908-S-5.1.

†Megaoryzomys curioi (ZFMK 2016-0981-sk), skull: www.morphdbase.de/?C_Koch_20200908-S-6.1.

## New Species Registration

The following information was supplied regarding the registration of a newly described species:

Publication LSID: urn:lsid:zoobank.org:pub:3E11AF88-BD56-40BE-9D43-EF6E5998E2D1.

Genus name:

Pattonimus gen. nov. LSID: urn:lsid:zoobank.org:act:83926983-C0A8-4337-B5F9-81B01CF7B487.

Species name:

Pattonimus ecominga sp. nov. LSID: urn:lsid:zoobank.org:act:15A88558-F671-46C8-8826-D0E3962F620C.

Pattonimus musseri sp. nov. LSID: urn:lsid:zoobank.org:act:A50ABD02-60BA-497C-9DCE-83D6C7811305.

## Supplemental Information

Supplemental information for this article can be found online at http://dx.doi.org/10.7717/peerj.10247#supplemental-information.

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
