# Peer review of "A new genus of oryzomyine rodents (Cricetidae, Sigmodontinae) with three new species from montane cloud forests, western Andean cordillera of Colombia and Ecuador"

_PeerJ, doi:10.7717/peerj.10247_

## Round 0.1 · original submission · Major Revisions

Reviewer 3 has suggested rejection and reviewer 2 made several suggestions. I would be willing to consider an improved version of the manuscript, especially addressing the reviewers' concerns.

·

Basic reporting

no comment

Experimental design

no comment

Validity of the findings

no comment

Additional comments

The article reports the discovery of a new oryzomyine genus obtained from field surveys in the Chocó ecoregion of Ecuador and Colombia. Brito et al. present an exhaustive and detailed anatomical description of the new genus and comparison with two closely related and a sympatric genus together with high-quality illustrations. From my point of view, the generic status is largely justified by phylogenetic analysis (molecular and morphological characters) and quantitative anatomical traits and dental morphology. I included minor suggestions in a marked pdf.

Reviewer 2 ·

Basic reporting

In this study the authors employ an integrative systematic approach and described a new genus within Oryzomyini, which has at least three new species, two of them are also named and diagnosed. For these descriptions and analyses, the authors perform an integrated approach from phylogenetic analyses, morphological comparisons and detailed anatomical evidence partially based on micro-computed tomography, on specimens colected in Ecuador and Colombia.

I think that the phylogenetic and descriptive approaches are very consistent and adequate to test the hypotheses about the differentiation and presence of novel clades.

English is clear, but I suggest to improve it in some parts of the manuscript.

Bibliographic references are appropriate.

Of course I have really enjoyed reading this study on Orizomini taxonomy and Neotropical biogeography.

Experimental design

Experimental design are relevant and meaninful.

In summary, the experimental design and conclusions are very robust. I only suggest a few questions from the statistical point of view, in relation to phylogenetic inference analyzes (all details are provided to the authors).

Validity of the findings

Findings of this manuscript are very interesting. I think that the conclusions of this article can have a great impact on what is the taxonomy and biogeography of these clades of small rodents. The approach is quite original. The discussion is very well structured and argued, with an easy and interesting reading.

Additional comments

Comments to the authors:

Phylogenetic and descriptive approaches in this study are very consistent and adequate to test the hypotheses about the differentiation and the presence of a new clade in this region.
The authors have made a very good comparison with the two closest Orizomini clades (Nephelomys and Mindomys), and in turn with the clade that geographically has the greatest phylogenetic similarities (Tanyuromys). I think the authors have correctly integrated both molecular data and descriptions of body and cranio-dental morphology, in order to make a good description of a new genus within Orizomini. Both phylogenetic analyzes and descriptions of external and cranio-dental morphology support the presence of a new genera within Orizomini. Also, the exclusive molar morphology of Pattonimus among oryzomyines provides the best evidence of the ecological uniqueness of this taxon.
The study is extensive and the descriptions of the study material are correct and very thorough. I consider it to be a very interesting contribution to the biogeography and taxonomy of rodents in one of the most diverse areas in the Neotropical region. The study contributes to expanding the taxonomy and knowledge of this large rodent clade. The manuscript has a very good balance between Tables and Figures on the main text, and those materials reported as Supplemental Information.

Of course I have really enjoyed reading this study on Orizomini taxonomy and Neotropical biogeography. One of the recommendations for future studies would be to explore the ecological spaces occupied by these new species based on ecological niche models.

In order to improve this manuscript, I have some major and minor recommendations to the authors. For instance, I think grammar and spelling need to be improved in many sections of this manuscript.

Major comments:

1) The authors performed PCAs in order to explore some differences in the putative species described here. In this sense, PCAs included six external, 19 cranial, and six dental morphological variables. Beyond that the logarithmized data tends to homogenize the variances in the data, I suggest (at least) to perform independent PCAs using variables corresponding to each category of data (e.g. PCA with only cranial measurements, or a OCA with only external measurements). By adjusting variables with such different magnitude values, the relative weight of a particular cranial-dental variable could be clouded. I think that this approach should be attempted, thinking about the significance of each variable in each main component.

2) I think that in many cases there are species with very incomplete sequences, such as Nephelomys devius, Pattonimus ecominga, Nephelomys keaysi, Pattonimus musseri (sequence data from S4). I think the authors should evaluate how this missing data within sequences might affect the phylogenetic inferences.

3) From lines 301 to 318, about the phylogenetic inference using the supermatrix. The authors use parsimony and argue that cytb mitochondrial DNA could be the reason for saturation at higher taxonomic level evaluated here.

I think there are many ways to assess the saturation level of a particular molecular marker, such as the use of the DAMBE program, which I recommend to be used by the authors. In this sense, and according to my proper experience in other rodent groups, the citb would not have to show high saturation at this taxonomic level ("tribe" level). On the contrary, I think that high levels of homoplasy could come from the morphological variation (e.g. how morphological variation has been treated in this study). In fact, it is very difficult to assign saturation problems to molecular markers when working together with sources of morphological variation. For this reason, I suggest to the authors to explain the possible limitations when both type of data sets (morphological variation vs. molecular variation) are taking into account in order to infer phylogenetic relationships.
Beyond these questions, it is clear from the phylogenetic analyzes (using both data sets and using only molecular data) that the clade proposed as a novel genus is recovered as monophyletic, reinforcing the authors' idea regarding the proposal of at least two new species.

4) Considering the phylogenetic inferences, did the authors use any substitution model for each independent molecular partition? Authors should estimate the particular mutation rates for each DNA fragment (nuclear and mitochondrial), and incorporate this information in the phylogenetic analyses (BI analysis, for instance). The steps would be to calculate the best nucleotide substitution model for each particular DNA fragment (e.g. using Jmodeltest), then concatenate the genes and check the level of saturation (e.g. DAMBE software), after that include this information into the Bayesian inference (e.g. BEAST program).

Minor corrections:

In this section I summarize minor concerns, which in many cases are style or writing suggestions.
Abstract:

1) About “Extensive field work in two protected areas enclosing outstanding remnants of Chocó montane forest retrieved a variety of small rodents”; this sentence sounds a bit ambiguous, I suggest explaining how the rodent diversity was found in the two protected areas (e.g. recovered a high diversity of small mammals).

2) I suggest to connect better this sentence “Colombian and Ecuadorian Pacific cloud forests are under rapid anthropic transformation”.

Introduction:

3) I suggest to replace this sentence “The Oryzomyini, measured by number of genera and species, is the largest tribe of the 56 sigmodontine radiation, and according to current counts it comprises about 152 living (including 57 those historical extinct) species distributed in 33 genera (Weksler, 2015;
Pardiñas et al., 2017)” by “The Oryzomyini is the largest tribe resulting from the large radiation of the 56 sigmodontine rodent clades, and according to current counts it comprises about 152 living (including 57 those historical extinct) species distributed in 33 genera (Weksler, 2015; Pardiñas et al., 2017)”.
I also suggest putting this tribe in a broader geographic context, and adding something like "It is also the tribe with the widest geographic distribution, extending from the southeastern United States of America to Tierra del Fuego and the Cape Horn islands, plus some oceanic islands and the Antillean region”.

4) I suggest to replace “For this reason, and because of its high degree of threats,…” by “For this reason, and because of its high degree of threats to biodiversity,…”.

Material and Methods:

5) This study only use pitfall traps… I ask the authors (and only as a particular concern) why sherman traps have not been used, which in addition to live captures of small cursorial mammals allow to have (to a much greater extent) captures of scansorial species (or inclusive arboreal rodents).

6) In the text, the section “Anatomy, age criteria and measurements” does not lead to its corresponding figure (can be?).

7) Line 181, about “Females and males were combined in all analyses, following Voss (1991) and Abreu-Jr. et al. 182 (2012), who concluded that sexual dimorphism was not an important source of morphometric variation in oryzomyine rodents”. I suppose this will depend on the type of specimens and their particular body weights. The most correct procedure in this case is to do an allometric analysis to evaluate possible sex differences in adults. I suggest justifying why the latter was not carried out.

8) In some parts of the text, the authors use the passive voice, such as “For the fresh tissue samples (90% ethanol), DNeasy (Qiagen) or Puregene 204 (Gentra) extraction kits were used. For museum samples the protocol of Bilton & Jaarola (1996) was used”. In this cases, I suggest to use the first person.

9) Line 201. I suggest to replace “We amplified two genes: the mitochondrial gene cytochrome-b (Cytb) using…” by “We amplified two genes: the mitochondrial cytochrome b (Cytb) gene using…”.

10) Line 207. Modify “gene, amplified using…” by “gene using…”.

11) Replaced “The amplicons were sequenced by Macrogen (South Korea, Inc)”.

12) Line 216. This sentence should be referred to the corresponding Table or Appendix: The taxonomic sampling of the used morphological matrix corresponds to that of Pine et al. (2012) with the additions of the new material described here.

13) Between lines 209 to 222. The authors refer to the morphological matrix in Supplemental Information S3, but this file corresponds to a nexus file from DNA sequences. Please, check carefully.

14) From lines 227 to 229. The authors should include de original number of sequences that correspond to this study and to these molecular partitions.

15) Line 236. Replace “…while the molecular only datasets were…” by “…while the molecular datasets were…”.

16) Lines 284-288. In relation to Figure 1, the authors report that Bayesian trees and maximum likelihood trees were similar in terms of clade conformation. In order to avoid the polytomies observed in the Bayesian tree, I suggest evaluating the possibility of reporting the maximum likelihood tree topology. In which case, reports the reason why the authors have not proceeded in this way.

17) Line 329. I suggest to replace “The reduced molecular only analysis (Fig. 2B) also recovered the new taxa as a monophyletic group, but sister to Mindomys hammondi” by “The reduced phylogenetic tree using only molecular data (Fig. 2B) also recovered the new taxa as a monophyletic clade, but sister to Mindomys hammondi”.

18) Fig. 2. The topology of the phylogenetic trees (A and B) corresponds to de BI or ML inference?

19) In order to improve this manuscript, suggest to the authors check carefully the grammar and spelling. English need to be improved in many sections of this manuscript.

20) Fig. 5. It is very confusing which letter corresponds to each specimen. Maybe there is something here that I did not understand very well. It is not understood to which occlusal series each taxon belongs. In addition to this, italicize the scientific names of the species.

21) Lines 404-407. This information seems somewhat redundant, consider whether to leave it or delete it.

22) Line 408-418. About the pelage description of each entity, authors say that specimens from Drácula and Río Manduriacu reserves are externally very similar, although the fur of latter is less dorso-ventrally counter shaded as they exhibit greyer bellies. These chromatic differences are also displayed in the tails, which are more dark above and below in animals from Río Manduriacu.
Is there any quantification regarding the population variability of these traits at the phenotype level? How many individuals have been compared in relation to these results? The authors should indicate in the text the number of individuals that have been taken into account to make these descriptions.

23) Lines 414-416. I suggest to rewrite this sentence as follows: In contrast, the antorbital bridge from specimens of Río Manduriacu is dorsally narrow, the alar fissure has a marked basal notch, and the lateral expansions of the parietals are absent.

24) Lines 447-450. Consider dividing this sentence into two parts.

Discussion:

25) From lines 880 to 882. Here, the authors say that this genus could represents a novel transition to a dental morphospace within the tribe, combining lamination, increased crown height (i.e., relatively more hypsodont), occlusal simplification, and a mesiodistally compressed m3 (Fig. 13).
I think it is difficult from these photographs to observe the hypsodoncia patterns. I suggest, if were possible, add some kind of scale.

26) Lines 922-924. Where the authros say “although an important degree of transverse lamination (i.e., confluence of directly opposing folds; Hershkovitz 1962: 92) is observed in a few taxa, such as the sigmodontine Irenomys”. There is any citation for the lamination pattern observed in Irenomys?

Legend of Tables and Figures:

Line 1444. I suggest to express in the following form: Clades are indicated by letters (A to D).
Line 1448. I suggest modify by “…from 45 terminals.”. It is already clear in the text the number of total base pairs in the concatenated DNA fragment.

Reviewer 3 ·

Basic reporting

Field sampling and curatorial efforts leading to these new specimens are commendable, as are the comparisons with relevant museum specimens across several collections and the careful morphological descriptions. Unfortunately, the molecular efforts are far more limited and essentially add new cyt b and IRBP data to the broader, existing datasets for these loci. The resulting support for the key clades (new genus, distinct species) is very limited. Statistical support is only moderate and ultimately based on these two loci.

The language and style of the manuscript are professional and only minor tweaking would be required from that standpoint.

Experimental design

The new molecular data are limited: 8 new IRBP sequences and 16 new cyt b sequences. Recognizing that these are the two most commonly used loci for phylogenetic analysis in Sigmodontinae, the resulting database is limited to support a new genus and three new species. It would seem that several additional loci would be required to secure such support.

In terms of taxonomic representation, the molecular data used to represent the genus Nephelomys are also very limited and make it difficult to assess whether these two genera are indeed reciprocally monophyletic or the new genus is embedded within Nephelomys.

Validity of the findings

The presentation of levels of divergence between the relevant taxa is very limited, both in the text and in figures and tables. Table 1 shows overall (presumably combined cytb-IRBP) divergence among genera, but the details are too limited to evaluate the meaning of these values. How does divergence in cyt b and IRBP compare to each other across species and genera? The pairwise comparisons among putative new species, provided in the text, are also sketchy and do not allow clear assessment of divergence and congruence across these two loci. The possibility that the proposed new genus is but part of Nephelomys, and the possibility that a single species is represented in the data cannot be critically assessed with the information provided by the authors.

Additional comments

I commend the field, curatorial, and museum research component of this project. However, I would encourage further efforts in securing additional representation of the relevant taxa (more specimens and taxa) and sequencing additional loci to further clarify the phylogenetic relations and distinctiveness of the taxa proposed here.

---

## Round 0.2 · Minor Revisions

I consider that you have addressed the major issues raised by the reviewers and I don't think the submission needs to be sent back to the reviewers.

Please make the following corrections (numbers indicate the lines in the reviewing PDF):
1: change “rodent” to “rodents”
32: change “the latter” to “them”
39: change “Biogeographic” to “biogeographic”
44: change “Province” to “province”
68: change “Biogeographic” to “biogeographic”
71: delete comma after “and”
72: change “Perú” to “Peru”
337: change “analyzes” to “analyses” and delete comma after “separately”
341: delete comma after “3)” and delete “it”
343: change “brother” to “sister taxon”
348: change “a brother” to “the sister taxon”
367: change “sized” to “sizes” and “rare” to “are”
440: change “brilliant” to “bright”
647: delete “of the generic name”
997: change “had long been“ to “have been long”
1001: change “was” to “were”
1021: change “appears” to “appear”

---

## Round 0.3 · accepted · Accept

Thanks for correcting your previous version of your manuscript.